# When Are Bias-Free ReLU Networks Effectively Linear Networks?

**Yedi Zhang**                                                                 *yedi@gatsby.ucl.ac.uk*
*Gatsby Computational Neuroscience Unit*
*University College London*

**Andrew Saxe**                                                                *a.saxe@ucl.ac.uk*
*Gatsby Computational Neuroscience Unit & Sainsbury Wellcome Centre*
*University College London*

**Peter E. Latham**                                                            *pel@gatsby.ucl.ac.uk*
*Gatsby Computational Neuroscience Unit*
*University College London*

**Reviewed on OpenReview:** *https://openreview.net/forum?id=Ucpfdn66k2*

## Abstract

We investigate the implications of removing bias in ReLU networks regarding their expressivity and learning dynamics. We first show that two-layer bias-free ReLU networks have limited expressivity: the only odd function two-layer bias-free ReLU networks can express is a linear one. We then show that, under symmetry conditions on the data, these networks have the same learning dynamics as linear networks. This enables us to give analytical time-course solutions to certain two-layer bias-free (leaky) ReLU networks outside the lazy learning regime. While deep bias-free ReLU networks are more expressive than their two-layer counterparts, they still share a number of similarities with deep linear networks. These similarities enable us to leverage insights from linear networks to understand certain ReLU networks. Overall, our results show that some properties previously established for bias-free ReLU networks arise due to equivalence to linear networks.

## 1 Introduction

Theorists make simplifications to real-world models because simplified models are mathematically more tractable, yet discoveries made in them may hold in general. For instance, linear models have illuminated benign overfitting (Bartlett et al., 2020) and double descent (Advani et al., 2020) in practical neural networks. In this paradigm, understanding the consequences of a simplification is critical, since it informs us which discoveries in simple models extend to complex ones. Here we inspect a specific simplification that is common in theoretical work on ReLU networks (Zhang et al., 2019; Du et al., 2019; Arora et al., 2019; Lyu & Li, 2020; Vardi & Shamir, 2021): the removal of the bias terms. The removal of bias not only appears in theoretical work but also has practical applications. Some real-world models adopt bias removal to introduce scale invariance, which can be a beneficial property for image denoising (Mohan et al., 2020; Zhang et al., 2022), image classification (Zarka et al., 2021), and diffusion models (Kadkhodaie et al., 2024). This paper seeks to illuminate the implications of bias removal in ReLU networks, and so provide insight for theorists on when bias removal is desirable.

We investigate how removing bias affects the expressivity and the learning dynamics of ReLU networks and identify scenarios where bias-free ReLU networks are effectively linear networks. For expressivity, we show that two-layer bias-free (leaky) ReLU networks cannot express odd functions except linear functions. This was proven for input uniformly distributed on a sphere (Basri et al., 2019, Theorem 2 and 4), but we prove it for arbitrary input with a simpler approach. We then consider deep bias-free (leaky) ReLU networks and

show a depth separation result, i.e., deep bias-free ReLU networks can express homogeneous nonlinear odd functions while two-layer ones cannot. For learning dynamics, we show that two-layer bias-free (leaky) ReLU networks have the same learning dynamics as a linear network when trained with square loss or logistic loss on symmetric datasets, whose target function is odd and input distribution is even. Our symmetry Condition 3 on the dataset incorporates the datasets studied in several prior works (Sarussi et al., 2021; Lyu et al., 2021; Zhang et al., 2024). We also present two cases where two-layer bias-free ReLU networks evolve like multiple independent linear networks. Finally, we empirically find that when the target function is linear, deep bias-free ReLU networks form low-rank weights similar to those in deep linear networks.

By revealing regimes where bias-free ReLU networks behave like linear networks, we provide an accessible way of understanding ReLU networks within these regimes, as well as a cautionary note that studying nonlinear behaviors generally requires stepping beyond these regimes. This understanding leverages insights from linear networks, which are simpler and thus enjoy much richer theoretical results than ReLU networks (Baldi & Hornik, 1989; Fukumizu, 1998; Saxe et al., 2014; 2019; Arora et al., 2018; Ji & Telgarsky, 2019; Lampinen & Ganguli, 2019; Gidel et al., 2019; Tarmoun et al., 2021; Braun et al., 2022; Ziyin et al., 2022). For example, we are able to give closed-form time-course solutions to certain two-layer ReLU networks outside the lazy learning regime in Corollary 9. Our findings suggest that the bias terms in a ReLU network play an important role in learning nonlinear tasks. Our contributions are the following:

- Section 3 proves the limited expressivity of bias-free (leaky) ReLU networks, and shows a depth separation result between two-layer and deep bias-free ReLU networks;

- Section 4.1 proves that under symmetry Condition 3 on the dataset, two-layer bias-free (leaky) ReLU networks trained with square loss or logistic loss evolve the same as linear networks, and gives analytical time-course solutions for ReLU networks in this regime.

- Section 4.2 shows that bias-free ReLU networks behave similarly to multiple independent linear networks on orthogonal and XOR datasets;

- Section 5 shows the similarities between deep bias-free ReLU networks and deep linear networks, and finds specific rank-one and rank-two structure in the weights.

## 1.1 Related Work

Basri et al. (2019) proved, using harmonic analysis, that two-layer bias-free ReLU networks can neither learn nor express odd nonlinear functions when input is uniformly distributed on a sphere (Basri et al., 2019, Theorem 2 and 4). We make a similar argument with a simpler proof. Our Theorem 1 handles arbitrary input, includes both ReLU and leaky ReLU networks, and the proof only involves rewriting the (leaky) ReLU activation function as the sum of a linear function and an absolute value function.

Lyu et al. (2021) proved two-layer bias-free leaky ReLU networks trained with logistic loss converge to a linear, max-margin classifier on linearly separable tasks with a data augmentation procedure. Our Theorem 8 shows that the learning dynamics of leaky ReLU networks in their setup is equivalent to that of a linear network. In light of this equivalence, their result is guaranteed given that linear networks trained with logistic loss converge to the max-margin classifier on linearly separable tasks (Soudry et al., 2018). In addition, we relax the assumption on the task from being linearly separable to being odd, and thus identify a practical challenge: the data augmentation procedure of Lyu et al. (2021) can cause the ReLU network to fail to learn a linearly non-separable task — a task the network might have succeeded to learn without data augmentation.

Zhang et al. (2024) found that two-layer bias-free ReLU networks have similar loss and weight norm curves as linear networks when trained on datasets with zero mean Gaussian input and a linear target. They reported that training the ReLU networks is about twice as slow as their linear counterpart. Our Theorem 8 explains their observation: we prove that the dynamics of two-layer bias-free ReLU networks is exactly twice as slow as their linear counterpart for a general class of datasets, including theirs.

A few other works have alluded to the connections between two-layer ReLU and linear networks. Sarussi et al. (2021) discovered that two-layer bias-free leaky ReLU networks converge to a decision boundary that is very close to linear when the teacher model is linear. Their theoretical results assume that the second

layer is fixed while we train all layers of the network. Saxe et al. (2022) studied gated deep linear networks and found they closely approximate a two-layer bias-free ReLU network trained on an XOR task. But they did not generalize the connection between gated linear networks and ReLU networks beyond the XOR case. Boursier & Flammarion (2024a) gave an example dataset with three scalar input data points, in which two-layer bias-free (leaky) ReLU networks converge to the linear, ordinary least square estimator. Holzmüller & Steinwart (2022) studied two-layer leaky ReLU networks with bias and found that they perform linear regression on certain data distributions with scalar input, because the bias fails to move far away from their initialization at zero.

While prior works have studied cases where bias-free ReLU networks behave like linear networks, their connections have not been explicitly highlighted or systematically summarized. This paper aims to bring the connections between bias-free ReLU and linear networks into focus, offering a comparative perspective on ReLU networks.

## 2 Preliminaries

**Notations**: Non-bold symbols denote scalars. Bold symbols denote vectors and matrices. Double-pipe brackets $\|\cdot\|$ denote the L2 norm of a vector or the Frobenius norm of a matrix. Angle brackets $\langle\cdot\rangle$ denote the average over the dataset. The circled dot $\odot$ denotes the element-wise product.

### 2.1 Two-Layer Bias-Free (Leaky) ReLU and Linear Networks

A two-layer bias-free (Leaky) ReLU network with $H$ hidden neurons is defined as

$$f(\boldsymbol{x}; \boldsymbol{W}) = \boldsymbol{W}_2\sigma(\boldsymbol{W}_1\boldsymbol{x}) = \sum_{h=1}^{H} w_{2h}\sigma\left(\boldsymbol{w}_{1h}^{\top}\boldsymbol{x}\right), \quad \text{where } \sigma(z) = \max(z, \alpha z), \alpha \in [0, 1]. \tag{1}$$

Here $\boldsymbol{x} \in \mathbb{R}^D$ is the input, $\boldsymbol{W}_1 \in \mathbb{R}^{H \times D}$ is the first-layer weight, $\boldsymbol{W}_2 \in \mathbb{R}^{1 \times H}$ is the second-layer weight, and $\boldsymbol{W}$ denotes all weights collectively. This is a ReLU network when $\alpha = 0$ and a leaky ReLU network when $\alpha \in (0, 1)$. When $\alpha = 1$, the network is a linear network, and can be written as $f(\boldsymbol{x}) = \boldsymbol{W}_2\boldsymbol{W}_1\boldsymbol{x}$. We also denote the linear network as $f^{\text{lin}}\left(\boldsymbol{x}; \boldsymbol{W}^{\text{lin}}\right) = \boldsymbol{W}_2^{\text{lin}}\boldsymbol{W}_1^{\text{lin}}\boldsymbol{x}$ when we need to distinguish it from ReLU networks.

We consider the rich regime (Woodworth et al., 2020) in which the network is initialized with small random weights. The network is trained with gradient descent on a dataset $\{\boldsymbol{x}_\mu, y_\mu\}_{\mu=1}^{P}$ consisting of $P$ samples. We study square loss $\mathcal{L} = \left\langle (y - f(\boldsymbol{x}))^2 \right\rangle / 2$ and logistic loss $\mathcal{L}_{\text{LG}} = \left\langle \ln\left(1 + e^{yf(\boldsymbol{x})}\right)\right\rangle$. We focus on square loss in the main text and provide derivations with logistic loss in the appendix. The learning rate is $\eta$ and the inverse of the learning rate is the time constant $\tau = 1/\eta$. In the limit of small learning rate, the gradient descent dynamics are well approximated by the gradient flow differential equations

$$\tau\dot{\boldsymbol{W}}_1 = \left\langle \sigma'(\boldsymbol{W}_1\boldsymbol{x}) \odot \boldsymbol{W}_2^{\top}\left(y - \boldsymbol{W}_2\sigma(\boldsymbol{W}_1\boldsymbol{x})\right)\boldsymbol{x}^{\top}\right\rangle, \tag{2a}$$

$$\tau\dot{\boldsymbol{W}}_2 = \left\langle \left(y - \boldsymbol{W}_2\sigma(\boldsymbol{W}_1\boldsymbol{x})\right)\sigma(\boldsymbol{W}_1\boldsymbol{x})^{\top}\right\rangle, \tag{2b}$$

where $\sigma'$ is the derivative of $\sigma$, $\odot$ is the element-wise product, and the angle brackets $\langle\cdot\rangle$ denote taking the average over the dataset.

For linear networks, $\sigma(z) = z$, the gradient flow dynamics can be written as

$$\tau\dot{\boldsymbol{W}}_1^{\text{lin}} = \boldsymbol{W}_2^{\text{lin}\top}\left(\boldsymbol{\beta}^{\top} - \boldsymbol{W}_2^{\text{lin}}\boldsymbol{W}_1^{\text{lin}}\boldsymbol{\Sigma}\right), \tag{3a}$$

$$\tau\dot{\boldsymbol{W}}_2^{\text{lin}} = \left(\boldsymbol{\beta}^{\top} - \boldsymbol{W}_2^{\text{lin}}\boldsymbol{W}_1^{\text{lin}}\boldsymbol{\Sigma}\right)\boldsymbol{W}_1^{\text{lin}\top}, \tag{3b}$$

where $\boldsymbol{\Sigma}$ denotes the input covariance and $\boldsymbol{\beta}$ denotes the input-output correlation,

$$\boldsymbol{\Sigma} = \left\langle \boldsymbol{x}\boldsymbol{x}^{\top}\right\rangle, \quad \boldsymbol{\beta} = \left\langle y\boldsymbol{x}\right\rangle. \tag{4}$$

## 2.2 Deep Networks

A deep neural network of depth $L$ is $f(\boldsymbol{x}) = h_L$ where $h_L$ is recursively defined as

$$
\begin{aligned}
\boldsymbol{h}_l &= \boldsymbol{W}_l \sigma(\boldsymbol{h}_{l-1}), \quad 2 \leq l \leq L, \\
\boldsymbol{h}_1 &= \boldsymbol{W}_1 \boldsymbol{x}.
\end{aligned}
\tag{5}
$$

Here $\boldsymbol{h}_1, \cdots, \boldsymbol{h}_{L-1}$ are vectors and $h_L$ is the scalar output. The gradient flow dynamics trained with square loss is

$$
\tau \dot{\boldsymbol{W}}_l = \left\langle \frac{\partial h_L}{\partial \boldsymbol{h}_l} (y - h_L) \sigma(\boldsymbol{h}_{l-1})^\top \right\rangle.
\tag{6}
$$

For deep linear networks, the gradient flow dynamics can be written as

$$
\tau \dot{\boldsymbol{W}}_l^{\mathrm{lin}} = \left( \prod_{i=l+1}^{L} \boldsymbol{W}_i^{\mathrm{lin}} \right)^\top \left( \boldsymbol{\beta}^\top - \prod_{i=1}^{L} \boldsymbol{W}_i^{\mathrm{lin}} \boldsymbol{\Sigma} \right) \left( \prod_{i=1}^{l-1} \boldsymbol{W}_i^{\mathrm{lin}} \right)^\top,
\tag{7}
$$

where $\prod_i \boldsymbol{W}_i$ represents the ordered product of matrices with the largest index on the left and smallest on the right.

## 3 Network Expressivity

We first examine the expressivity of bias-free ReLU networks. It is well known that standard ReLU networks with bias are universal approximators (Hornik et al., 1989; Pinkus, 1999) while bias-free ReLU networks are not since they can only express positively homogeneous functions, i.e., $g(a\boldsymbol{x}) = ag(\boldsymbol{x}) \forall a > 0$. Moreover, Section 3.1 shows that two-layer bias-free ReLU networks cannot express any odd function except linear functions. Section 3.2 shows that deep bias-free ReLU networks are more expressive than two-layer ones, but are still limited to positively homogeneous functions.

### 3.1 Two-Layer Bias-Free (Leaky) ReLU Networks

**Theorem 1.** *The set of functions that can be expressed by two-layer bias-free (leaky) ReLU networks is a subset of the set of functions of the form: $f(\boldsymbol{x}) = h(\boldsymbol{x}) + g(\boldsymbol{x})$, where $h(\boldsymbol{x})$ is linear and $g(\boldsymbol{x})$ is a positively homogeneous even function, meaning $g(\boldsymbol{x}) = g(-\boldsymbol{x})$ and $g(a\boldsymbol{x}) = ag(\boldsymbol{x}) \forall a > 0$.*

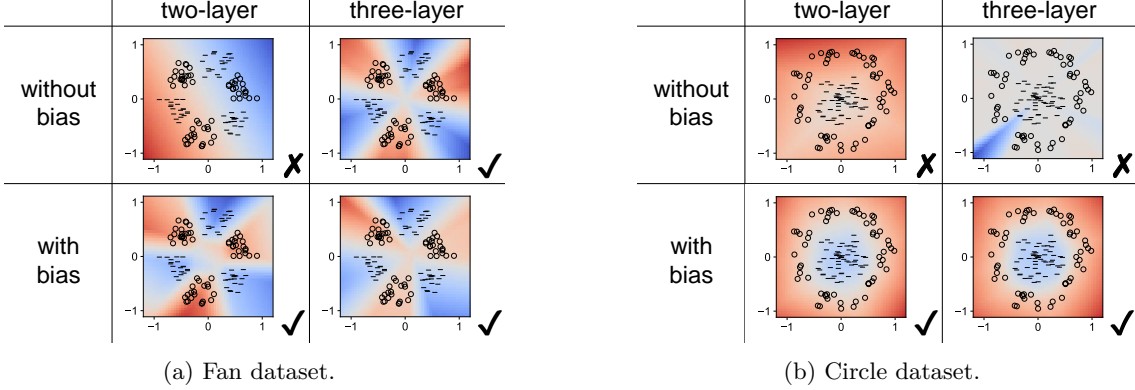

(a) Fan dataset.    (b) Circle dataset.

Figure 1: The expressivity of two-layer and deep ReLU networks with and without bias. The networks are trained with logistic loss until the loss stops decreasing. The empty circles are data points with $+1$ labels; short lines are data points with $-1$ labels. The network output is plotted in color. (a) The fan dataset is odd, homogeneous, and satisfies Condition 3. Two-layer bias-free ReLU networks cannot express it. (b) The circle dataset is not homogeneous. Two-layer and deep bias-free ReLU networks cannot express it. Experimental details are provided in Appendix H.

*Proof.* Notice that the (leaky) ReLU activation function admits a decomposition[1]: $\sigma(z) = \frac{1+\alpha}{2}z + \frac{1-\alpha}{2}|z|$. Thus, any two-layer (leaky) ReLU network can be written as

$$f(\boldsymbol{x}; \boldsymbol{W}) = \sum_{h=1}^{H} w_{2h}\sigma(\boldsymbol{w}_{1h}\boldsymbol{x}) = \sum_{h=1}^{H} w_{2h} \left[ \frac{1+\alpha}{2}\boldsymbol{w}_{1h}\boldsymbol{x} + \frac{1-\alpha}{2}|\boldsymbol{w}_{1h}\boldsymbol{x}| \right], \tag{8}$$

which is a linear function plus a positively homogeneous even function. □

**Corollary 2.** *The only odd function that bias-free two-layer (leaky) ReLU networks can express is the linear function.*

Due to this restricted expressivity, two-layer bias-free ReLU networks fail to classify the fan dataset, as shown in Figure 1a.

### 3.2 Deep Bias-Free (Leaky) ReLU Networks

Similarly to two-layer bias-free ReLU networks, deep bias-free ReLU networks can express only positively homogeneous functions. Thus, as shown in Figure 1b, both two-layer and deep bias-free ReLU networks fail to classify the circle dataset. However, in contrast to two-layer bias-free ReLU networks, deep bias-free ReLU networks can express some odd nonlinear functions. For instance, for two-dimensional input $\boldsymbol{x} = [x_1, x_2]^\top$, the function below is odd, nonlinear, and can be implemented by a three-layer bias-free ReLU network,

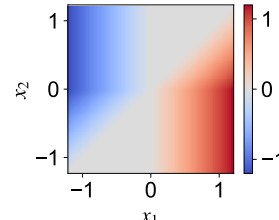

$$g(\boldsymbol{x}) = \sigma(\sigma(x_1) - \sigma(x_2)) - \sigma(\sigma(-x_1) - \sigma(-x_2)), \quad \text{where } \sigma(z) = \max(z, 0). \tag{9}$$

Figure 2: Function $g(\boldsymbol{x})$ defined in Equation (9) is plotted with color.

Thus, we have a depth separation result for bias-free ReLU networks: there exist odd nonlinear functions, such as $g(\boldsymbol{x})$ defined in Equation (9) and visualized in Figure 2, that two-layer bias-free ReLU networks cannot express but deep bias-free ReLU networks can.

## 4 Learning Dynamics in Two-Layer Bias-Free ReLU Networks

### 4.1 Symmetric Datasets

Section 3.1 has proven that two-layer bias-free ReLU networks cannot express odd functions except linear functions. We now show that under Condition 3 on the dataset, two-layer bias-free ReLU networks not only find a linear solution but also have the same learning dynamics as a two-layer linear network.

**Condition 3.** The dataset satisfies the following two symmetry conditions:

1. The empirical input data distribution is even: $p(\boldsymbol{x}) = p(-\boldsymbol{x})$;

2. The target function is odd: $y(\boldsymbol{x}) = -y(-\boldsymbol{x})$.

**Remark 4.** For infinite data, the first part of Condition 3 includes common distributions, such as any Gaussian distribution with zero mean. For finite data, the first part of Condition 3 means that if $\boldsymbol{x}$ is present in the dataset, $-\boldsymbol{x}$ is also present. Condition 3 includes the dataset studied in Lyu et al. (2021). They considered linearly separable binary classification tasks with a data augmentation procedure in which $(-\boldsymbol{x}, -y)$ is added to the dataset if $(\boldsymbol{x}, y)$ is in the dataset. We have the same assumption on the input data distribution but relax the assumption on the target function from being linearly separable to being odd.

The key implication of Condition 3 is that the input covariance matrix and the input-output correlation averaged over any half space are equal to those averaged over the entire space. We state this in Lemma 5 and prove it in Appendix B.2.

---

[1]We note that decomposing the ReLU activation function into a linear and an even function is technically uncomplicated and has appeared in prior literature before, e.g., Ghorbani et al. (2021, Section 1.3) and Martinelli et al. (2024, page 4).

**Lemma 5.** *Let set $\mathbb{S}^+$ be an arbitrary half space divided by a hyperplane with normal vector $\boldsymbol{r}$, namely $\mathbb{S}^+ = \{\boldsymbol{x} \in \mathbb{R}^D | \boldsymbol{r}^\top \boldsymbol{x} > 0\}$. Under Condition 3, we have $\forall \boldsymbol{r}$*

$$\langle \boldsymbol{x}\boldsymbol{x}^\top \rangle_{\mathbb{S}^+} = \boldsymbol{\Sigma}, \quad \langle \boldsymbol{x}y(\boldsymbol{x}) \rangle_{\mathbb{S}^+} = \boldsymbol{\beta}. \tag{10}$$

*Recall that $\boldsymbol{\Sigma}$ and $\boldsymbol{\beta}$ are averages over the entire space as defined in Equation (4).*

Under Condition 3, two-layer bias-free (leaky) ReLU networks initialized with small random weights evolve approximately according to a linear differential equation in the early phase of learning. We can solve the approximate linear differential equation and bound the errors of the approximation, leading to the following lemma.

**Lemma 6.** *We define the initialization scale as $w_{\text{init}} = \max(\|\boldsymbol{W}_1(0)\|, \|\boldsymbol{W}_2(0)\|)$. For time $t < \frac{\tau}{s + \text{Tr}\,\boldsymbol{\Sigma}} \ln \frac{1}{w_{\text{init}}}$, the solution to the dynamics of two-layer (leaky) ReLU networks starting from small initialization exhibits exponential growth along one direction with small errors*

$$\boldsymbol{W}_1(t) = e^{\frac{\alpha+1}{2\tau} st} \boldsymbol{r}_1 \bar{\boldsymbol{\beta}}^\top + O(w_{\text{init}}), \quad \boldsymbol{W}_2(t) = e^{\frac{\alpha+1}{2\tau} st} \boldsymbol{r}_1^\top + O(w_{\text{init}}). \tag{11}$$

*where $s = \|\boldsymbol{\beta}\|, \bar{\boldsymbol{\beta}} = \boldsymbol{\beta}/s$, and $\boldsymbol{r}_1$ is determined by random initialization $\boldsymbol{r}_1 = \left(\boldsymbol{W}_1(0)\bar{\boldsymbol{\beta}} + \boldsymbol{W}_2^\top(0)\right)/2$.*

The proof for Lemma 6 can be found in Appendix C.1. Lemma 6 indicates that the weights of the ReLU network and the linear network form the same rank-one structure in the early phase, differing only in the speed of exponential growth. At the end of the early phase, the weights are aligned and rank-one with bounded errors, $O(w_{\text{init}})$. For simplicity, we will assume in Assumption 7 that the weights are exactly aligned and rank-one, which is justified when the initialization is infinitesimal, $w_{\text{init}} \to 0$. Assumption 7 further assumes that $\boldsymbol{W}_2$ has equal L2 norms for their positive and negative elements. This assumption is supported by the fact that $\boldsymbol{W}_2$ is proportional to $\boldsymbol{r}_1$, which follows a zero-mean Gaussian distribution because the initial weights $\boldsymbol{W}_1(0), \boldsymbol{W}_2(0)$ are sampled from a zero-mean Gaussian distribution. As the network width approaches infinity, $\boldsymbol{W}_2$ consists of infinitely many zero-mean Gaussian random samples, which have equal L2 norms for their positive and negative elements.

**Assumption 7.** At initialization, there exists an unit vector $\boldsymbol{r}$ such that $\boldsymbol{W}_1 = \boldsymbol{W}_2^\top \boldsymbol{r}^\top$, and the L2 norms of the positive and negative elements in $\boldsymbol{W}_2$ are equal, that is $\|\max(\boldsymbol{W}_2, 0)\| = \|\max(-\boldsymbol{W}_2, 0)\|$ where $\max(\cdot)$ is applied element-wise.

**Theorem 8.** *A two-layer (leaky) ReLU network and a linear network are trained with square or logistic loss starting from weights which differ by a scale factor, $\boldsymbol{W}(0) = \sqrt{2/(\alpha+1)}\,\boldsymbol{W}^{\text{lin}}(0)$. Under Condition 3 on the dataset and Assumption 7 on the initial weights, the learning dynamics of the two-layer (leaky) ReLU network reduces to*

$$\tau \dot{\boldsymbol{W}}_1 = \frac{\alpha+1}{2} \boldsymbol{W}_2^\top \boldsymbol{\beta}^\top - \left(\frac{\alpha+1}{2}\right)^2 \boldsymbol{W}_2^\top \boldsymbol{W}_2 \boldsymbol{W}_1 \boldsymbol{\Sigma}, \tag{12a}$$

$$\tau \dot{\boldsymbol{W}}_2 = \frac{\alpha+1}{2} \boldsymbol{\beta}^\top \boldsymbol{W}_1^\top - \left(\frac{\alpha+1}{2}\right)^2 \boldsymbol{W}_2 \boldsymbol{W}_1 \boldsymbol{\Sigma} \boldsymbol{W}_1^\top. \tag{12b}$$

*For all $t \geq 0$, Assumption 7 remains valid and the following hold:*

*1. The (leaky) ReLU network implements the same linear function as the linear network with scaled time*

$$f(\boldsymbol{x}; \boldsymbol{W}(t)) = f^{\text{lin}}\left(\boldsymbol{x}; \boldsymbol{W}^{\text{lin}}\left(\frac{\alpha+1}{2}t\right)\right); \tag{13}$$

*2. The weights in the (leaky) ReLU network are the same as scaled weights in the linear network*

$$\boldsymbol{W}(t) = \sqrt{\frac{2}{\alpha+1}} \boldsymbol{W}^{\text{lin}}\left(\frac{\alpha+1}{2}t\right). \tag{14}$$

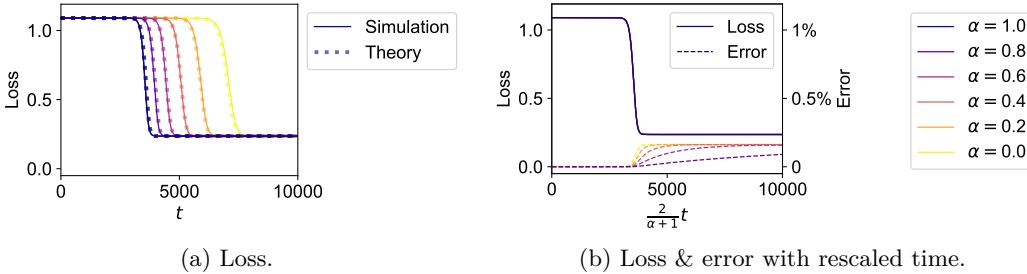

(a) Loss.  (b) Loss & error with rescaled time.

Figure 3: Two-layer bias-free (leaky) ReLU networks can evolve like a linear network. (a) Loss curves with different leaky ReLU parameters $\alpha$ (note $\alpha = 1$ is a linear network). The simulations match the theoretical solutions in Equation (15). The loss converges to global minimum, which is not zero due to the restricted expressivity of two-layer bias-free ReLU networks. (b) The simulated loss curves are plotted against a rescaled time axis; they collapse to one curve, demonstrating the (leaky) ReLU and linear networks are implementing the same linear function as in Equation (13). The error, defined as $\left\| \sqrt{\frac{\alpha+1}{2}} \boldsymbol{W} \left( \frac{2}{\alpha+1} t \right) - \boldsymbol{W}^{\mathrm{lin}}(t) \right\| / \left\| \boldsymbol{W}^{\mathrm{lin}}(t) \right\|$, is less than 0.3%, demonstrating that the weights in the (leaky) ReLU network are close to the weights in the linear network as in Equation (14). The errors are not exactly zero because the initial weights are sampled from a zero-mean Gaussian distribution, which does not satisfy Assumption 7 but better reflects practical initialization schemes. Experimental details are provided in Appendix H.

The proof for Theorem 8 can be found in Appendix C.2. The key step is that the reduced dynamics of the ReLU network given in Equation (12) is the same as that for the linear network in Equation (3), modulo the constant coefficients. Hence, apart from the fact that learning is $(\alpha + 1)/2$ times slower and the weights are $\sqrt{2/(\alpha+1)}$ times larger, the ReLU network has the same learning dynamics as its linear counterpart.

We validate Theorem 8 and the plausibility of Assumption 7 with numerical simulations in Figure 3. In Figure 3b, the initialization is small random Gaussian weights and thus does not satisfy Assumption 7, yet Theorem 8 holds with small errors (less than 0.3%). Furthermore, we provide theoretical proof that Theorem 8 holds with L2 regularization and empirical evidence that some of Theorem 8 hold with large initialization and a moderately large learning rate in Appendices C.4 to C.6.

If the input covariance is white, we can further write down the exact time-course solution in closed form for two-layer bias-free (leaky) ReLU networks by adopting the solutions from linear networks (Braun et al., 2022, Theorem 3.1). This gives us the following corollary.

**Corollary 9.** *For learning with square loss, if the input covariance is white, $\boldsymbol{\Sigma} = \boldsymbol{I}$, the solution to Equation (13) is $f(\boldsymbol{x}; \boldsymbol{W}) = \boldsymbol{w}(t)^{\top} \boldsymbol{x}$ with*

$$\boldsymbol{w}(t) = \left( 1 + \frac{q_1}{q_2} e^{-2s\tilde{t}} \right) \left[ \bar{\boldsymbol{\beta}} \left( 1 - \frac{q_1}{q_2} e^{-2s\tilde{t}} \right) + \frac{2}{q_2} \left( \boldsymbol{I} - \bar{\boldsymbol{\beta}} \bar{\boldsymbol{\beta}}^{\top} \right) \boldsymbol{r} e^{-s\tilde{t}} \right]$$

$$\left[ \frac{4}{q_2^2} \left( w_{\mathrm{init}}^{-2} + \left( 1 - \left( \boldsymbol{r}^{\top} \bar{\boldsymbol{\beta}} \right)^2 \right) \tilde{t} \right) e^{-2s\tilde{t}} + \frac{1}{s} \left( 1 + \frac{q_1^2}{q_2^2} e^{-2s\tilde{t}} \right) \left( 1 - e^{-2s\tilde{t}} \right) \right]^{-1}, \qquad (15)$$

*where $\tilde{t}$ is a shorthand for rescaled time $\tilde{t} = \frac{\alpha+1}{2\tau} t$ and the constant quantities are $s = \|\boldsymbol{\beta}\|, \bar{\boldsymbol{\beta}} = \boldsymbol{\beta}/s, q_1 = 1 - \boldsymbol{r}^{\top} \bar{\boldsymbol{\beta}}, q_2 = 1 + \boldsymbol{r}^{\top} \bar{\boldsymbol{\beta}}, w_{\mathrm{init}} = \|\boldsymbol{W}_1(0)\|$.*

The solution given in Equation (15) matches simulations, as shown in Figure 3a.

Since the time evolution of two-layer bias-free (leaky) ReLU networks is the same as that of linear networks (modulo scale factors), their converged weights will also be the same. For learning with square loss, linear networks converge to the ordinary least squares solution (Saxe et al., 2014). For linearly separable binary classification with logistic loss, linear networks converge to the max-margin (hard margin SVM) solution (Soudry et al., 2018). Thus two-layer bias-free (leaky) ReLU networks also converge to these solutions when they behave like linear networks; see Appendix C.3.

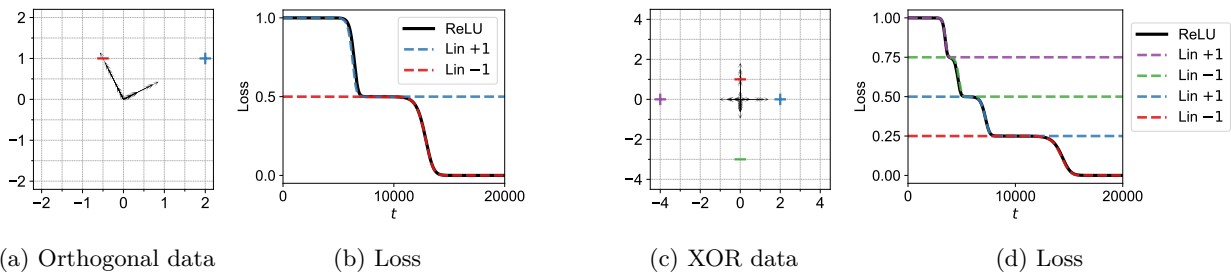

(a) Orthogonal data      (b) Loss      (c) XOR data      (d) Loss

Figure 4: Two-layer bias-free ReLU networks can evolve like multiple independent linear networks. (a) An orthogonal input dataset used in (Boursier et al., 2022, Figure 3). The $+$ and $-$ signs represent data points with $+1$ and $-1$ labels respectively. Their different colors are used only to distinguish the loss curves. The black arrows are the first-layer weights at convergence. (b) The loss curve of the ReLU network overlaps with two linear networks trained on each of the two data points respectively. (c) An XOR-like dataset. (d) The loss curve of the ReLU network overlaps with four linear networks trained on each of the four data points separately. Details: We use summed (instead of averaged) square loss for this figure. The initial losses are vertically aligned to help illustrate the overlap. More details are in Appendix H.

**Corollary 10.** *Under the same conditions as Theorem 8, the two-layer bias-free (leaky) ReLU network converges to a linear solution $f(\boldsymbol{x}; \boldsymbol{W}(\infty)) = \boldsymbol{w}^{*\top}\boldsymbol{x}$. For square loss, $\boldsymbol{w}^*$ is the ordinary least squares solution, $\boldsymbol{w}^* = \boldsymbol{\Sigma}^{-1}\boldsymbol{\beta}$, which is the global minimum. For linearly separable binary classification with logistic loss, $\boldsymbol{w}^*$ aligns with the max-margin solution, $\boldsymbol{w}^*/\|\boldsymbol{w}^*\| = \boldsymbol{w}_{\mathrm{svm}}/\|\boldsymbol{w}_{\mathrm{svm}}\|$ where*

$$\boldsymbol{w}_{\mathrm{svm}} = \operatorname*{argmin}_{\boldsymbol{w}}\|\boldsymbol{w}\|^2 \quad \text{s.t.} \quad y_\mu \boldsymbol{w}^\top \boldsymbol{x}_\mu \geq 1, \ \forall \ \mu = 1, \cdots, P. \tag{16}$$

## 4.2 Orthogonal and XOR Datasets

In Section 4.1, we showed that for symmetric datasets satisfying Condition 3, a two-layer bias-free (leaky) ReLU network evolves like a linear network. For more general datasets, the equivalence no longer holds, but comparing ReLU with linear networks remains useful for understanding the learning dynamics of ReLU networks. Specifically, we highlight two cases where a two-layer bias-free ReLU network evolves like multiple independent linear networks. These cases, i.e., an orthogonal input dataset and an XOR-like dataset, are commonly considered in theoretical literature on ReLU network learning dynamics, while their connection to linear network dynamics has not been previously highlighted.

We first look into datasets with orthogonal input, that is $\forall \mu \neq \nu, \boldsymbol{x}_\mu^\top \boldsymbol{x}_\nu = 0$, a common setting in the literature (Boursier et al., 2022; Telgarsky, 2023; Frei et al., 2023b;c; Kou et al., 2023; Dana et al., 2025). We illustrate with a dataset with two orthogonal data points from Boursier et al. (2022), and handle an arbitrary number of data points in Proposition 18. We train a two-layer bias-free ReLU network on this dataset (Figure 4a), and reproduce the loss curve in (Boursier et al., 2022, Figure 3). We then train two two-layer linear networks on each data point separately. We find that the timing and the amount of the loss drop overlap with the loss curves of the two linear networks as shown in Figure 4b. To explain this overlap, we plot the first-layer weights of the ReLU network in black arrows in Figure 4a and find that the weights align with either one of the two data points. Since the two directions are orthogonal, the learning dynamics of the two groups of neurons decouple, as derived in Proposition 18. Each group of neurons evolves like a linear network trained on that single data point. Hence, weights in the ReLU network evolve like a linear network trained on either one of the two data points separately. For a dataset with an arbitrary number of orthogonal inputs, the dynamics of the two-layer bias-free ReLU network evolves like two linear networks trained separately on data points with positive labels and data points with negative labels, as validated in Figure 9. The same applies to learning with logistic loss, as shown in Figure 8.

We observe similar behaviors in the XOR-like task shown in Figure 4c. XOR-like datasets are also a common setting in the literature (Refinetti et al., 2021; Saxe et al., 2022; Frei et al., 2023a; Meng et al., 2024; Xu et al., 2024; Glasgow, 2024). As shown in Figure 4d, we find that the loss curves of a two-layer bias-free ReLU

network trained on the XOR task overlap with four linear networks trained on each data point separately. In this case, the dynamics of multiple linear networks well approximate that of a ReLU network, even though the ReLU network learns a nonlinear function.

In Figures 4b and 4d, the loss curves go through multiple drops, each corresponding to learning a data point. Similar behaviors were examined by Boursier et al. (2022); Xu et al. (2024) and characterized as saddle-to-saddle dynamics. The connections we find between ReLU and linear networks may help understand these behaviors in ReLU networks because saddle-to-saddle dynamics has been well studied for linear networks (Saxe et al., 2014; 2019; Gissin et al., 2020; Jacot et al., 2022; Berthier, 2023; Pesme & Flammarion, 2023).

## 5 Learning Dynamics in Deep Bias-Free ReLU Networks

In Section 4, we showed that two-layer bias-free ReLU networks behave like linear networks under symmetry Condition 3 and small initialization. This does not extend to deep bias-free networks. When trained on a dataset satisfying Condition 3, deep bias-free ReLU networks can learn nonlinear solutions if the target function is nonlinear, as shown in Figure 1a (upper right). Nonetheless, we find deep bias-free ReLU networks can form low-rank weights that are similar to those in deep linear networks. We give an example where the empirical input distribution is even and the target function is linear.

In a deep linear network, weights form an approximately rank-one structure and adjacent layers are approximately aligned when trained from small initialization (Ji & Telgarsky, 2019; Advani et al., 2020; Atanasov et al., 2022; Marion & Chizat, 2024). The rank-one weight matrices can be written approximately as outer-products of two vectors

$$\boldsymbol{W}_1^{\text{lin}} = u\boldsymbol{r}_1\boldsymbol{r}^\top = u\begin{bmatrix}\boldsymbol{r}_1^+ \\ \boldsymbol{r}_1^-\end{bmatrix}\boldsymbol{r}^\top, \tag{17a}$$

$$\boldsymbol{W}_l^{\text{lin}} = u\boldsymbol{r}_l\boldsymbol{r}_{l-1}^\top = u\begin{bmatrix}\boldsymbol{r}_l^+\boldsymbol{r}_{l-1}^{+\top} & \boldsymbol{r}_l^+\boldsymbol{r}_{l-1}^{-\top} \\ \boldsymbol{r}_l^-\boldsymbol{r}_{l-1}^{+\top} & \boldsymbol{r}_l^-\boldsymbol{r}_{l-1}^{-\top}\end{bmatrix}, \quad l = 2, \cdots, L-1, \tag{17b}$$

$$\boldsymbol{W}_L^{\text{lin}} = u\boldsymbol{r}_{L-1}^\top = u\begin{bmatrix}\boldsymbol{r}_{L-1}^{+\top} & \boldsymbol{r}_{L-1}^{-\top}\end{bmatrix}, \tag{17c}$$

where $u > 0$ represents the norm of each layer, and $\boldsymbol{r}, \boldsymbol{r}_1, \boldsymbol{r}_2, \cdots, \boldsymbol{r}_L$ are unit norm column vectors. The vectors $\boldsymbol{r}_l^+, \boldsymbol{r}_l^-$ denote the positive and negative elements in $\boldsymbol{r}_l$. The equal norm $u$ of all layers is a consequence of small initialization (Du et al., 2018). Note that the weights can be written in blocks, as Equation (17), only after permuting the positive and negative elements. We use this permuted notation for the sake of exposition; no additional assumptions are required.

In a deep ReLU network, we empirically find that when the weights are trained from small initialization and the target function is linear, the weights exhibit a particular rank-one and rank-two structure, as shown in Figures 5 and 10. For the first and last layers, the weights in the deep ReLU network have the same rank-one structure as their linear counterpart. For the intermediate layers, weights in the deep ReLU network are rank-two. Specifically, positive weights in the ReLU network correspond to positive weights in the linear network and zero weights in the ReLU network correspond to negative weights in the linear network. Based on the empirical observation, we propose the following conjecture on the weights of the deep ReLU network.

**Conjecture 11.** A deep bias-free ReLU network is trained from small initial weights on a dataset where the empirical input distribution is even, $p(\boldsymbol{x}) = p(-\boldsymbol{x})$, and the target function is linear, that is the target output is generated as $y = \boldsymbol{w}^{*\top}\boldsymbol{x}$. We conjecture that the weights at a certain time $t_0$ during training take the following form:

$$\boldsymbol{W}_1 = u\boldsymbol{r}_1\boldsymbol{r}^\top = u\begin{bmatrix}\boldsymbol{r}_1^+ \\ \boldsymbol{r}_1^-\end{bmatrix}\boldsymbol{r}^\top, \tag{18a}$$

$$\boldsymbol{W}_l = u\begin{bmatrix}\sqrt{2}\boldsymbol{r}_l^+\boldsymbol{r}_{l-1}^{+\top} & \boldsymbol{0} \\ \boldsymbol{0} & \sqrt{2}\boldsymbol{r}_l^-\boldsymbol{r}_{l-1}^{-\top}\end{bmatrix}, \quad l = 2, \cdots, L-1, \tag{18b}$$

$$\boldsymbol{W}_L = u\boldsymbol{r}_{L-1}^\top = u\begin{bmatrix}\boldsymbol{r}_{L-1}^{+\top} & \boldsymbol{r}_{L-1}^{-\top}\end{bmatrix}, \tag{18c}$$

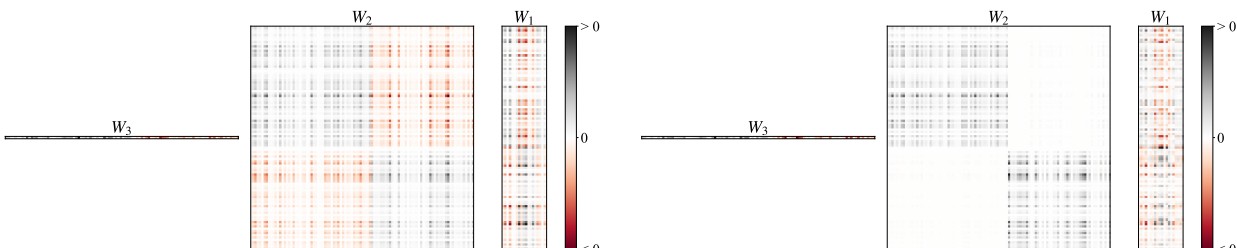

(a) Weights in a 3-layer linear network as Equation (17). (b) Weights in a 3-layer ReLU network as Equation (18).

Figure 5: Low-rank weights in deep linear and ReLU bias-free networks. A three-layer linear network and a three-layer ReLU network are trained on the same dataset starting from the same small random weights. The dataset has a linear target function and an even empirical input data distribution. We plot the weights when the loss has approached zero. $\boldsymbol{W}_1$, $\boldsymbol{W}_3$, and positive elements in $\boldsymbol{W}_2$ have approximately the same structure in the linear and ReLU networks. Elements of $\boldsymbol{W}_2$ that are negative in the linear network are approximately zero in the ReLU network. The neurons are permuted for visualization. Simulations with deeper networks are presented in Figure 10. Experimental details are provided in Appendix H.

where the notations are consistent with Equation (17). We also conjecture that $\|\boldsymbol{r}_l^+\| = \|\boldsymbol{r}_l^-\|$, $l = 1, 2, \cdots, L-1$.

**Proposition 12.** *If Conjecture 11 is true, then for all $t \geq t_0$, the weights of the deep bias-free ReLU network will maintain the form in Equation* (18) *and the network implements a linear function given by*

$$f(\boldsymbol{x}; \boldsymbol{W}) = \boldsymbol{W}_L \boldsymbol{W}_{L-1} \cdots \boldsymbol{W}_2 \sigma(\boldsymbol{W}_1 \boldsymbol{x}) = \frac{1}{2} \boldsymbol{W}_L \cdots \boldsymbol{W}_2 \boldsymbol{W}_1 \boldsymbol{x}. \tag{19}$$

*Moreover, its learning dynamics is described by*

$$\tau \dot{\boldsymbol{W}}_l = \left( \prod_{l'=l+1}^{L} \boldsymbol{W}_{l'} \right)^{\top} \left( \frac{1}{2} \boldsymbol{\beta}^{\top} - \frac{1}{4} \prod_{l'=1}^{L} \boldsymbol{W}_{l'} \boldsymbol{\Sigma} \right) \left( \prod_{l'=l-1}^{L} \boldsymbol{W}_{l'} \right)^{\top}, \tag{20}$$

*which is the same as that of a deep linear network in Equation* (7), *modulo the constant coefficients.*

We provide the proof of Proposition 12 in Appendix E and offer a brief explanation for Equation (19) here. In the first equality of Equation (19), we dropped the activation functions except for the one between the first and second layers. This is because the second layer weights, $\boldsymbol{W}_2$, is non-negative as shown in Figure 5b, and so is the output of a ReLU activation function, $\sigma(\boldsymbol{W}_1 \boldsymbol{x})$. Hence, their product, $\boldsymbol{W}_2 \sigma(\boldsymbol{W}_1 \boldsymbol{x})$, is also non-negative. We thus have $\sigma(\boldsymbol{W}_2 \sigma(\boldsymbol{W}_1 \boldsymbol{x})) = \boldsymbol{W}_2 \sigma(\boldsymbol{W}_1 \boldsymbol{x})$. The same applies to all subsequent layers. The second equality is obtained by substituting the weights defined in Equation (18) into the expression.

While the proof of Conjecture 11 remains an open question, we provide some intuition to interpret it. Specifically, we notice that in deep linear networks and certain deep ReLU networks, the weights of an intermediate layer align with the inputs to that layer. For example, the second-layer weight in a deep linear network is $\boldsymbol{W}_2^{\text{lin}} = u \boldsymbol{r}_2 \boldsymbol{r}_1^{\top}$ as given in Equation (17). Every row of $\boldsymbol{W}_2^{\text{lin}}$ aligns with $\boldsymbol{r}_1^{\top}$, which is parallel to any input to the second layer, $\boldsymbol{W}_1^{\text{lin}} \boldsymbol{x} = u \boldsymbol{r}_1 \boldsymbol{r}^{\top} \boldsymbol{x}$. The second-layer weight in the deep ReLU network is given in Equation (18). Some rows of $\boldsymbol{W}_2$ align with $\begin{bmatrix} \boldsymbol{r}_1^{+\top} & \boldsymbol{0} \end{bmatrix}$, which is parallel to some inputs ($\boldsymbol{r}^{\top} \boldsymbol{x} > 0$) to the second layer, $\sigma(\boldsymbol{W}_1 \boldsymbol{x}) = u \begin{bmatrix} \boldsymbol{r}_1^+ \\ \boldsymbol{0} \end{bmatrix} \boldsymbol{r}^{\top} \boldsymbol{x}$. Other rows of $\boldsymbol{W}_2$ align with $\begin{bmatrix} \boldsymbol{0} & \boldsymbol{r}_1^{-\top} \end{bmatrix}$, which is parallel with inputs ($\boldsymbol{r}^{\top} \boldsymbol{x} < 0$) to the second layer, $\sigma(\boldsymbol{W}_1 \boldsymbol{x}) = u \begin{bmatrix} \boldsymbol{0} \\ \boldsymbol{r}_1^- \end{bmatrix} \boldsymbol{r}^{\top} \boldsymbol{x}$. This alignment phenomenon has been proven for deep linear networks (Ji & Telgarsky, 2019; Marion & Chizat, 2024), and we here find similar phenomena empirically in deep ReLU networks.

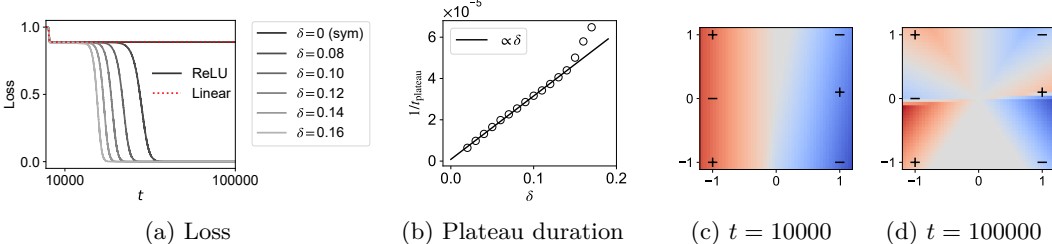

(a) Loss      (b) Plateau duration      (c) $t = 10000$      (d) $t = 100000$

Figure 6: Two-layer bias-free linear/ReLU network trained on a dataset that slightly violates the symmetry Condition 3. The $+$ and $-$ signs represent data points with $+1$ and $-$ labels respectively. The right middle data point is slight asymmetric with input coordinates $(1, \delta)$. (a) Loss curves of the ReLU network with different $\delta$ and the linear network with $\delta = 0.1$. (b) The duration of the plateau, during which the ReLU network implements a nearly linear solution, scales approximately with $1/\delta$. (c,d) The ReLU network output during and at the end of training. This ReLU network is trained on the dataset with $\delta = 0.1$. Experimental details are provided in Appendix H.

## 6 Discussion

### 6.1 Implication of Bias Removal

We studied the implications of removing bias in ReLU networks in terms of the expressivity and learning dynamics. Theorem 1 shows that two-layer bias-free (leaky) ReLU networks cannot express any odd functions except for linear functions. Theorem 8 shows that for datasets with an even input distribution and an odd target function, two-layer bias-free (leaky) ReLU networks have the same time evolution as a linear network (modulo scale factors) under initialization Assumption 7. We also presented examples in which the bias-free ReLU network evolves like multiple independent linear networks, in Section 4.2. In these cases, comparing a bias-free ReLU network with its linear counterpart provides an intuitive understanding of the behavior of ReLU networks. On the flip side, the simplicity of bias-free ReLU networks suggests that ReLU networks with bias may exhibit more complicated behaviors, which are not fully addressed by studies on bias-free networks, and remain open questions.

One common argument in studies of bias-free ReLU networks is that we can stack the input $\boldsymbol{x}$ with an additional one, i.e., $\tilde{\boldsymbol{x}} = [\boldsymbol{x}, 1]$. The behaviors of a biased ReLU network with input $\boldsymbol{x}$ are the same as those of a bias-free ReLU network with input $\tilde{\boldsymbol{x}}$, suggesting that the implication of bias removal might be trivial. While this argument is valid in certain settings (Allen-Zhu et al., 2019; Zou et al., 2020), it comes with important caveats in others. For example, Soudry et al. (2018) showed that two-layer bias-free ReLU networks trained with logistic loss converge to the max-margin classifier on linearly separable datasets. As clarified by Soudry et al. (2018), this technical result holds when the inputs are stacked with an additional one. However, the max-margin solution of the dataset $\{\tilde{\boldsymbol{x}}_\mu, y_\mu\}_{\mu=1}^P$ is not the max-margin solution of the original dataset $\{\boldsymbol{x}_\mu, y_\mu\}_{\mu=1}^P$. Hence, a ReLU network with bias converges to a solution different from the max-margin solution obtained by the bias-free ReLU network. This distinction highlights that the presence of bias terms can change the inductive bias of the ReLU network, leading to convergence to different solutions.

### 6.2 Perturbed Symmetric Dataset

We have shown an exact equivalence between two-layer bias-free (leaky) ReLU networks and linear networks under symmetry Condition 3 on the dataset in Theorem 8. In practice, no datasets satisfies Condition 3 precisely. However, two-layer bias-free ReLU networks may still struggle to fit a dataset that approximately satisfies Condition 3. We present a simple example with six data points in Figure 6. The ReLU network loss curve closely follows the linear network loss curve in the early phase, when it first learns a nearly linear solution, as shown in Figures 6a and 6c. After a plateau, the ReLU network diverges from the linear network dynamics and converges to a nonlinear solution. The more symmetric the dataset, the longer the plateau a two-layer bias-free ReLU network undergoes before learning a nonlinear solution. When the dataset is

exactly symmetric, the ReLU network never learns a nonlinear solution. As shown in Figure 6b, the inverse of the plateau duration scales approximately linearly with the deviation of the asymmetric data point, $\delta$. The scaling becomes less accurate for larger $\delta$ because the corresponding dataset more severely violates symmetric Condition 3, where the ReLU network no longer behaves like a linear network. Furthermore, as shown in Figure 6d, the decision boundaries at convergence are close to the data points, and thus probably not robust.

We note a concurrent work (Boursier & Flammarion, 2024b) that considers two-layer bias-free ReLU networks trained on symmetric datasets with a different form of perturbation. They assume the empirical input distribution is even and the target output is generated as $y = \boldsymbol{w}^{*\top}\boldsymbol{x} + \xi$, where $\xi$ is the noise sampled independently from a zero-mean Gaussian distribution. In their setup, the ReLU network converges to the ordinary least square linear estimator when the number of training samples exceeds a certain threshold. In their case, the mean of the noise approaches zero as the number of training samples increases, which reduces the asymmetric perturbation and leads to convergence to a linear solution. By analogy, in our case, a smaller $\delta$ means a smaller asymmetric perturbation, which leads to a longer plateau during which the ReLU network is stuck at a linear solution.

## Acknowledgement

The authors are grateful to Peter Orbanz, Ingo Steinwart, Spencer Frei, Rodrigo Carrasco-Davis, Lukas Braun, Clémentine Dominé, Zheng He, and William Dorrell for helpful discussions. The authors thank the following funding sources: Gatsby Charitable Foundation (GAT3850) to YZ, AS, and PEL; Sainsbury Wellcome Centre Core Grant from Wellcome (219627/Z/19/Z) to AS; Schmidt Science Polymath Award to AS; Wellcome Trust (110114/Z/15/Z) to PEL.

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

## A  Additional Related Work

**Implicit / Simplicity Bias.** Many works have studied the implicit bias or simplicity bias of two-layer bias-free ReLU networks under various assumptions on the dataset. Brutzkus et al. (2018); Wang et al. (2019); Lyu et al. (2021); Sarussi et al. (2021); Wang & Ma (2023) considered linearly separable binary classification tasks. Phuong & Lampert (2021); Wang & Pilanci (2022); Min et al. (2024) studied orthogonally separable classification (i.e., where for every pair of labeled examples $(\boldsymbol{x}_i, y_i), (\boldsymbol{x}_j, y_j)$ we have $\boldsymbol{x}_i^\top \boldsymbol{x}_j > 0$ if $y_i = y_j$ and $\boldsymbol{x}_i^\top \boldsymbol{x}_j \leq 0$ if otherwise). Boursier et al. (2022); Frei et al. (2023b;c); Kou et al. (2023); Dana et al. (2025) studied binary classification with exactly or nearly orthogonal input (i.e., where $\boldsymbol{x}_i^\top \boldsymbol{x}_j = 0$ if $i \neq j$). Orthogonal input is a sufficient condition for linear separability for binary classification tasks. Refinetti et al. (2021); Frei et al. (2023a); Meng et al. (2024); Xu et al. (2024) studied XOR-like datasets. Vardi et al. (2022); Frei et al. (2023d) studied datasets with adversarial noise. We add to this line of research by studying a case with extreme simplicity bias, i.e., behaving like linear networks.

**Low-Rank Weights.** Maennel et al. (2018) is the seminal work on the low-rank weights in two-layer ReLU networks trained from small initialization. They described the phenomenon as "quantizing", where the first layer weight vectors align with a small number of directions in the early phase of training. Luo et al. (2021) identified when two-layer bias-free ReLU networks form low-rank weights in terms of the initialization and the network width. Timor et al. (2023) provided cases where gradient flow on two-layer and deep ReLU networks provably minimize or not minimize the ranks of weight matrices. Frei et al. (2023c); Kou et al. (2023) computed the numerical rank of the converged weights of two-layer bias-free ReLU networks for nearly orthogonal datasets, and found that weights in leaky ReLU networks have rank at most two and weights in ReLU networks have a numerical rank upper bounded by a constant. Chistikov et al. (2023) showed two-layer bias-free ReLU networks are implicitly biased to learn the network of minimal rank under the assumption that training points are correlated with the teacher neuron. Min et al. (2024); Boursier & Flammarion (2024a) studied the early phase learning dynamics to understand how the low-rank weights form. Petrini et al. (2022; 2023) conducted experiments on practical datasets to show that two-layer bias-free ReLU networks learn sparse features, which can be detrimental and lead to overfitting. Le & Jegelka (2022) generalize the low-rank phenomenon in linear and ReLU networks to arbitrary non-homogeneous networks whose last few layers contain linear fully-connected and linear ResNet blocks.

## B  Useful Lemmas

### B.1  Grönwall's Inequality

Grönwall's Inequality (Grönwall, 1919) is a common tool to obtain error bounds when considering approximate differential equations.

**Lemma 13** (Grönwall's Inequality). *Let $I$ denote an interval of the real line of the form $[a, \infty)$ or $[a, b]$ or $[a, b)$ with $a < b$. Let $\alpha, \beta$ and $u$ be real-valued functions defined on $I$. Assume that $\beta$ and $u$ are continuous and that the negative part of $\alpha$ is integrable on every closed and bounded subinterval of $I$. If $\beta$ is non-negative and $u$ satisfies the integral inequality*

$$u(t) \leq \alpha(t) + \int_a^t \beta(s)u(s)ds, \quad \forall t \in I,$$

*then*

$$u(t) \leq \alpha(t) + \int_a^t \alpha(s)\beta(s)e^{\int_s^t \beta(r)dr}ds, \quad t \in I. \tag{21}$$

### B.2  Data Statistics

We here prove Lemma 5, which states the input covariance matrix and the input-output correlation averaged over any half space are equal to those averaged over the entire space, under Condition 3.

*Proof of Lemma 5.* Define $\mathbb{S}^- = \{\boldsymbol{x} \in \mathbb{R}^D | \boldsymbol{r}^\top \boldsymbol{x} < 0\}$. Because Condition 3 states that $p(\boldsymbol{x})$ is even, we have

$$\int_{\mathbb{S}^+} p(\boldsymbol{x})d\boldsymbol{x} = \int_{\mathbb{S}^-} p(\boldsymbol{x})d\boldsymbol{x} = \frac{1}{2}.$$

By the definition of the conditional expectation, we have that

$$\left\langle \boldsymbol{x}\boldsymbol{x}^\top \right\rangle_{\mathbb{S}^+} \equiv \frac{\int_{\mathbb{S}^+} \boldsymbol{x}\boldsymbol{x}^\top p(\boldsymbol{x})d\boldsymbol{x}}{\int_{\mathbb{S}^+} p(\boldsymbol{x})d\boldsymbol{x}} = 2\int_{\mathbb{S}^+} \boldsymbol{x}\boldsymbol{x}^\top p(\boldsymbol{x})d\boldsymbol{x},$$

$$\left\langle \boldsymbol{x}y(\boldsymbol{x}) \right\rangle_{\mathbb{S}^+} \equiv \frac{\int_{\mathbb{S}^+} \boldsymbol{x}y(\boldsymbol{x})p(\boldsymbol{x})d\boldsymbol{x}}{\int_{\mathbb{S}^+} p(\boldsymbol{x})d\boldsymbol{x}} = 2\int_{\mathbb{S}^+} \boldsymbol{x}y(\boldsymbol{x})p(\boldsymbol{x})d\boldsymbol{x}.$$

Because $\boldsymbol{x}\boldsymbol{x}^\top p(\boldsymbol{x})$ is an even function about $\boldsymbol{x}$, the integral of $\boldsymbol{x}\boldsymbol{x}^\top p(\boldsymbol{x})$ over $\mathbb{S}^+$ or $\mathbb{S}^-$ is the same

$$\int_{\mathbb{S}^+} \boldsymbol{x}\boldsymbol{x}^\top p(\boldsymbol{x})d\boldsymbol{x} = \int_{\mathbb{S}^-} \boldsymbol{x}\boldsymbol{x}^\top p(\boldsymbol{x})d\boldsymbol{x}.$$

Thus, the average of $\boldsymbol{x}\boldsymbol{x}^\top$ over $\mathbb{S}^+$ is equal to the average in the entire space,

$$\boldsymbol{\Sigma} = \int_{\mathbb{S}^+} \boldsymbol{x}\boldsymbol{x}^\top p(\boldsymbol{x})d\boldsymbol{x} + \int_{\mathbb{S}^-} \boldsymbol{x}\boldsymbol{x}^\top p(\boldsymbol{x})d\boldsymbol{x} = 2\int_{\mathbb{S}^+} \boldsymbol{x}\boldsymbol{x}^\top p(\boldsymbol{x})d\boldsymbol{x} = \left\langle \boldsymbol{x}\boldsymbol{x}^\top \right\rangle_{\mathbb{S}^+}. \tag{22}$$

Because $\boldsymbol{x}y(\boldsymbol{x})$ is also an even function about $\boldsymbol{x}$ under Condition 3, the same argument holds

$$\boldsymbol{\beta} = \int_{\mathbb{S}^+} \boldsymbol{x}y(\boldsymbol{x})p(\boldsymbol{x})d\boldsymbol{x} + \int_{\mathbb{S}^-} \boldsymbol{x}y(\boldsymbol{x})p(\boldsymbol{x})d\boldsymbol{x} = 2\int_{\mathbb{S}^+} \boldsymbol{x}y(\boldsymbol{x})p(\boldsymbol{x})d\boldsymbol{x} = \left\langle \boldsymbol{x}y(\boldsymbol{x}) \right\rangle_{\mathbb{S}^+}. \tag{23}$$

Note that Equation (23) needs both conditions in Condition 3, that are $p(\boldsymbol{x}) = p(-\boldsymbol{x})$ and $y(\boldsymbol{x}) = -y(-\boldsymbol{x})$. However, Equation (22) only needs the first condition, that is $p(\boldsymbol{x}) = p(-\boldsymbol{x})$. □

**Lemma 14.** *Under Condition 3, the first terms in the differential Equation* (2) *reduce to*

$$\left\langle \sigma'(\boldsymbol{W}_1\boldsymbol{x}) \odot \boldsymbol{W}_2^\top y\boldsymbol{x}^\top \right\rangle = \frac{\alpha + 1}{2}\boldsymbol{W}_2^\top \boldsymbol{\beta}^\top, \tag{24a}$$

$$\left\langle y\sigma(\boldsymbol{W}_1\boldsymbol{x})^\top \right\rangle = \frac{\alpha + 1}{2}\boldsymbol{\beta}^\top \boldsymbol{W}_1^\top. \tag{24b}$$

*Proof.* Let us consider the $h$-th row of the matrix $\left\langle \sigma'(\boldsymbol{W}_1\boldsymbol{x}) \odot \boldsymbol{W}_2^\top y\boldsymbol{x}^\top \right\rangle$, which is

$$\left\langle \sigma'(\boldsymbol{w}_{1h}\boldsymbol{x})w_{2h}y\boldsymbol{x}^\top \right\rangle = \frac{1}{2}\left\langle \alpha w_{2h}y\boldsymbol{x}^\top \right\rangle_{\boldsymbol{w}_{1h}\boldsymbol{x}<0} + \frac{1}{2}\left\langle w_{2h}y\boldsymbol{x}^\top \right\rangle_{\boldsymbol{w}_{1h}\boldsymbol{x}>0} = \frac{\alpha + 1}{2}w_{2h}\boldsymbol{\beta}^\top,$$

where the first equality is the law of total expectation and the second equality uses Lemma 5. Because the same holds for all rows, Equation (24a) is true.

Let us consider the $h$-th element of the row vector $\left\langle y\sigma(\boldsymbol{W}_1\boldsymbol{x})^\top \right\rangle$, which is

$$\left\langle y\sigma(\boldsymbol{w}_{1h}\boldsymbol{x}) \right\rangle = \frac{1}{2}\left\langle \alpha y\boldsymbol{w}_{1h}\boldsymbol{x} \right\rangle_{\boldsymbol{w}_{1h}\boldsymbol{x}<0} + \frac{1}{2}\left\langle y\boldsymbol{w}_{1h}\boldsymbol{x} \right\rangle_{\boldsymbol{w}_{1h}\boldsymbol{x}>0} = \frac{\alpha + 1}{2}\boldsymbol{w}_{1h}\boldsymbol{\beta},$$

where the first equality is the law of total expectation and the second equality uses Lemma 5. Because the same holds for all elements, Equation (24b) is true. □

**Lemma 15.** *The second terms in the differential Equation* (2) *can be bounded by the norm of the weights and the trace of the input covariance matrix.*

1. $\left\| \left\langle \sigma(\boldsymbol{W}_1\boldsymbol{x})\sigma(\boldsymbol{W}_1\boldsymbol{x})^\top \right\rangle \right\| \le \|\boldsymbol{W}_1\|^2 \operatorname{Tr}\boldsymbol{\Sigma}$.

2. $\left\| \left\langle \sigma'(\boldsymbol{W}_1\boldsymbol{x}) \odot \boldsymbol{W}_2^\top \boldsymbol{W}_2\sigma(\boldsymbol{W}_1\boldsymbol{x})\boldsymbol{x}^\top \right\rangle \right\| \le \|\boldsymbol{W}_2\|^2\|\boldsymbol{W}_1\| \operatorname{Tr}\boldsymbol{\Sigma}$.

*Proof.* For the first inequality,

$$\left\|\left\langle\sigma(\boldsymbol{W}_1\boldsymbol{x})\sigma(\boldsymbol{W}_1\boldsymbol{x})^\top\right\rangle\right\| \leq \left\langle\|\sigma(\boldsymbol{W}_1\boldsymbol{x})\|^2\right\rangle$$
$$\leq \left\langle\|\boldsymbol{W}_1\boldsymbol{x}\|^2\right\rangle$$
$$\leq \left\langle\|\boldsymbol{W}_1\|^2\|\boldsymbol{x}\|^2\right\rangle$$
$$= \|\boldsymbol{W}_1\|^2\operatorname{Tr}\boldsymbol{\Sigma}.$$

For the second inequality,

$$\left\|\left\langle\sigma'(\boldsymbol{W}_1\boldsymbol{x})\odot\boldsymbol{W}_2^\top\boldsymbol{W}_2\sigma(\boldsymbol{W}_1\boldsymbol{x})\boldsymbol{x}^\top\right\rangle\right\| \leq \left\|\left\langle\boldsymbol{W}_2^\top\boldsymbol{W}_2\sigma(\boldsymbol{W}_1\boldsymbol{x})\boldsymbol{x}^\top\right\rangle\right\|$$
$$\leq \|\boldsymbol{W}_2\|^2\left\langle\|\boldsymbol{W}_1\boldsymbol{x}\|\|\boldsymbol{x}\|\right\rangle$$
$$\leq \|\boldsymbol{W}_2\|^2\left\langle\|\boldsymbol{W}_1\|\|\boldsymbol{x}\|^2\right\rangle$$
$$= \|\boldsymbol{W}_2\|^2\|\boldsymbol{W}_1\|\operatorname{Tr}\boldsymbol{\Sigma}.$$

$\square$

## C  Two-Layer Networks on Symmetric Datasets

### C.1  Learning Dynamics: Early Phase

In the early phase of learning, the network output is small compared to the target output because the initialization is small. Lemma 16 specifies how small the norm of the weights is.

**Lemma 16.** *Denote the larger L2 norm of the weights in a two-layer network as $u(t) = \max\{\|\boldsymbol{W}_1(t)\|, \|\boldsymbol{W}_2(t)\|\}$. The initial weights are small, that is $w_{\mathrm{init}} \equiv u(0) \ll 1$. For two-layer linear, ReLU, or leaky ReLU networks trained with square loss from small initialization, $u(t)$ is bounded by*

$$u(t) \leq u(0)e^{(s+\operatorname{Tr}\boldsymbol{\Sigma})t/\tau}, \tag{25}$$

*for time $t < \frac{\tau}{s+\operatorname{Tr}\boldsymbol{\Sigma}}\ln\frac{1}{w_{\mathrm{init}}}$.*

*Proof.* For two-layer linear, ReLU, or leaky ReLU networks, the learning dynamics are given in general in Equation (2). Using the equality in Lemma 14 and the inequality in Lemma 15, we can bound the dynamics of $u^2$ as

$$\tau\frac{d}{dt}u^2 = \tau\frac{d}{dt}\|\boldsymbol{W}_2\|^2 = (\alpha+1)\boldsymbol{\beta}^\top\boldsymbol{W}_1^\top\boldsymbol{W}_2^\top - 2\boldsymbol{W}_2\left\langle\sigma(\boldsymbol{W}_1\boldsymbol{x})\sigma(\boldsymbol{W}_1\boldsymbol{x})^\top\right\rangle\boldsymbol{W}_2^\top$$
$$\leq \left|(\alpha+1)\boldsymbol{\beta}^\top\boldsymbol{W}_1^\top\boldsymbol{W}_2^\top\right| + \left|2\boldsymbol{W}_2\left\langle\sigma(\boldsymbol{W}_1\boldsymbol{x})\sigma(\boldsymbol{W}_1\boldsymbol{x})^\top\right\rangle\boldsymbol{W}_2^\top\right|$$
$$\leq 2\|\boldsymbol{\beta}\|\|\boldsymbol{W}_1\|\|\boldsymbol{W}_2\| + 2\|\boldsymbol{W}_2\|^2\|\boldsymbol{W}_1\|^2\operatorname{Tr}\boldsymbol{\Sigma}$$
$$\leq 2su^2 + 2u^4\operatorname{Tr}\boldsymbol{\Sigma}.$$

where $s = \|\boldsymbol{\beta}\|$. For $u < 1$, we have

$$\tau\frac{d}{dt}u^2 \leq 2su^2 + 2u^4\operatorname{Tr}\boldsymbol{\Sigma} < 2\left(s+\operatorname{Tr}\boldsymbol{\Sigma}\right)u^2.$$

Via Lemma 13 Grönwall's Inequality, we obtain

$$u^2 \leq w_{\mathrm{init}}^2 e^{2(s+\operatorname{Tr}\boldsymbol{\Sigma})t/\tau} \quad \Rightarrow \quad u(t) \leq w_{\mathrm{init}}e^{(s+\operatorname{Tr}\boldsymbol{\Sigma})t/\tau}.$$

This holds for

$$t < \frac{\tau}{s+\operatorname{Tr}\boldsymbol{\Sigma}}\ln\frac{1}{w_{\mathrm{init}}}.$$

$\square$

Since the weights are small in the early phase, we can approximate the early phase dynamics with only the first terms in Equation (2), that is

$$\tau \dot{\boldsymbol{W}}_1 \approx \left\langle \sigma'(\boldsymbol{W}_1 \boldsymbol{x}) \odot \boldsymbol{W}_2^\top y \boldsymbol{x}^\top \right\rangle = \frac{\alpha+1}{2} \boldsymbol{W}_2^\top \boldsymbol{\beta}^\top, \tag{26}$$

$$\tau \dot{\boldsymbol{W}}_2 \approx \left\langle y \sigma(\boldsymbol{W}_1 \boldsymbol{x})^\top \right\rangle = \frac{\alpha+1}{2} \boldsymbol{\beta}^\top \boldsymbol{W}_1^\top, \tag{27}$$

where the equalities hold under Condition 3 as proven by Lemma 14. For Lemma 6, we solve the approximate early phase dynamics and show that the errors introduced by the approximation are bounded. We presented Lemma 6 in the main text and now prove it.

*Proof of Lemma 6.* We first consider the approximate learning dynamics:

$$\tau \dot{\widetilde{\boldsymbol{W}}}_1 = \frac{\alpha+1}{2} \widetilde{\boldsymbol{W}}_2^\top \boldsymbol{\beta}^\top, \quad \tau \dot{\widetilde{\boldsymbol{W}}}_2 = \frac{\alpha+1}{2} \boldsymbol{\beta}^\top \widetilde{\boldsymbol{W}}_1^\top. \tag{28}$$

This is a linear dynamical system with an analytical solution available. We re-write it as:

$$\tau \frac{d}{dt} \widetilde{\boldsymbol{W}} = \frac{\alpha+1}{2} \boldsymbol{M} \widetilde{\boldsymbol{W}}, \quad \text{where } \boldsymbol{M} = \begin{bmatrix} \boldsymbol{0} & \boldsymbol{\beta} \\ \boldsymbol{\beta}^\top & 0 \end{bmatrix}, \widetilde{\boldsymbol{W}} = \begin{bmatrix} \widetilde{\boldsymbol{W}}_1^\top \\ \widetilde{\boldsymbol{W}}_2 \end{bmatrix}. \tag{29}$$

Since matrix $\boldsymbol{M}$ only has two nonzero eigenvalues $\pm s$, the solution to Equation (29) is

$$\begin{aligned}
\widetilde{\boldsymbol{W}}(t) = {} & \frac{1}{2} e^{\frac{\alpha+1}{2\tau} st} \begin{bmatrix} \bar{\boldsymbol{\beta}} \\ 1 \end{bmatrix} \left( \bar{\boldsymbol{\beta}}^\top \boldsymbol{W}_1^\top(0) + \boldsymbol{W}_2(0) \right) \\
& + \frac{1}{2} e^{-\frac{\alpha+1}{2\tau} st} \begin{bmatrix} \bar{\boldsymbol{\beta}} \\ -1 \end{bmatrix} \left( \bar{\boldsymbol{\beta}}^\top \boldsymbol{W}_1^\top(0) - \boldsymbol{W}_2(0) \right) + \begin{bmatrix} \left( \boldsymbol{I} - \bar{\boldsymbol{\beta}} \bar{\boldsymbol{\beta}}^\top \right) \boldsymbol{W}_1^\top(0) \\ 0 \end{bmatrix}.
\end{aligned} \tag{30}$$

Note that only the first term in Equation (30) is growing.

We then consider the exact learning dynamics given by Equation (2) and prove its solution is close to $\widetilde{\boldsymbol{W}}(t)$. The dynamics of the difference between the exact and approximate dynamics are

$$\tau \frac{d}{dt} \left( \widetilde{\boldsymbol{W}}_1 - \boldsymbol{W}_1 \right) = \frac{\alpha+1}{2} \left( \widetilde{\boldsymbol{W}}_2 - \boldsymbol{W}_2 \right)^\top \boldsymbol{\beta}^\top + \left\langle \sigma'(\boldsymbol{W}_1 \boldsymbol{x}) \odot \boldsymbol{W}_2^\top \boldsymbol{W}_2 \sigma(\boldsymbol{W}_1 \boldsymbol{x}) \boldsymbol{x}^\top \right\rangle \tag{31a}$$

$$\tau \frac{d}{dt} \left( \widetilde{\boldsymbol{W}}_2 - \boldsymbol{W}_2 \right) = \frac{\alpha+1}{2} \boldsymbol{\beta}^\top \left( \widetilde{\boldsymbol{W}}_1 - \boldsymbol{W}_1 \right)^\top + \boldsymbol{W}_2 \left\langle \sigma(\boldsymbol{W}_1 \boldsymbol{x}) \sigma(\boldsymbol{W}_1 \boldsymbol{x})^\top \right\rangle. \tag{31b}$$

We re-write Equation (31) as

$$\tau \frac{d}{dt} \delta \boldsymbol{W} = \frac{\alpha+1}{2} \boldsymbol{M} \delta \boldsymbol{W} + \boldsymbol{\epsilon}, \tag{32}$$

The norm of the two components of $\boldsymbol{\epsilon}$ can be bounded via Lemma 15

$$\left\| \left\langle \sigma'(\boldsymbol{W}_1 \boldsymbol{x}) \odot \boldsymbol{W}_2^\top \boldsymbol{W}_2 \sigma(\boldsymbol{W}_1 \boldsymbol{x}) \boldsymbol{x}^\top \right\rangle \right\| \leq u^3 \operatorname{Tr} \boldsymbol{\Sigma},$$

$$\left\| \boldsymbol{W}_2 \left\langle \sigma(\boldsymbol{W}_1 \boldsymbol{x}) \sigma(\boldsymbol{W}_1 \boldsymbol{x})^\top \right\rangle \right\| \leq u^3 \operatorname{Tr} \boldsymbol{\Sigma}.$$

We can then substitute in Equation (25) and obtain

$$\|\boldsymbol{\epsilon}\| \leq \sqrt{2} u^3 \operatorname{Tr} \boldsymbol{\Sigma} < \sqrt{2} u_0^3 e^{3(s+\operatorname{Tr} \boldsymbol{\Sigma})t/\tau} \operatorname{Tr} \boldsymbol{\Sigma}.$$

We now bound the norm of $\boldsymbol{W} - \widetilde{\boldsymbol{W}}$

$$
\begin{aligned}
\left\| \boldsymbol{W} - \widetilde{\boldsymbol{W}} \right\| &= \left\| \int_0^t \frac{\alpha+1}{2} \boldsymbol{M} \left( \boldsymbol{W} - \widetilde{\boldsymbol{W}} \right) + \boldsymbol{\epsilon} dt \right\| \\
&\leq \int_0^t \| \boldsymbol{M} \| \left\| \boldsymbol{W} - \widetilde{\boldsymbol{W}} \right\| + \| \boldsymbol{\epsilon} \| dt \\
&\leq \int_0^t \left( \sqrt{2} s \left\| \boldsymbol{W} - \widetilde{\boldsymbol{W}} \right\| + \sqrt{2} u_0^3 e^{3(s + \operatorname{Tr} \boldsymbol{\Sigma}) t / \tau} \operatorname{Tr} \boldsymbol{\Sigma} \right) dt \\
&\leq \frac{\sqrt{2} u_0^3 \operatorname{Tr} \boldsymbol{\Sigma}}{3(s + \operatorname{Tr} \boldsymbol{\Sigma})} \left( e^{3(s + \operatorname{Tr} \boldsymbol{\Sigma}) t / \tau} - 1 \right) + \sqrt{2} s \int_0^t \left\| \boldsymbol{W} - \widetilde{\boldsymbol{W}} \right\| dt.
\end{aligned}
$$

Via Lemma 13 Grönwall's Inequality, we obtain

$$
\begin{aligned}
\left\| \boldsymbol{W} - \widetilde{\boldsymbol{W}} \right\| &\leq \frac{\sqrt{2} \operatorname{Tr} \boldsymbol{\Sigma} u_0^3}{3(s + \operatorname{Tr} \boldsymbol{\Sigma})} \left[ e^{3(s + \operatorname{Tr} \boldsymbol{\Sigma}) t / \tau} - 1 + \int_0^t \left( e^{3(s + \operatorname{Tr} \boldsymbol{\Sigma}) t' / \tau} - 1 \right) \sqrt{2} s e^{\sqrt{2} s t'} dt' \right] \\
&= \frac{\sqrt{2} \operatorname{Tr} \boldsymbol{\Sigma} u_0^3}{3(s + \operatorname{Tr} \boldsymbol{\Sigma})} \left[ e^{3(s + \operatorname{Tr} \boldsymbol{\Sigma}) t / \tau} + \frac{\sqrt{2} s \left( e^{[3(s + \operatorname{Tr} \boldsymbol{\Sigma}) + \sqrt{2} s] t / \tau} - 1 \right)}{3(s + \operatorname{Tr} \boldsymbol{\Sigma}) + \sqrt{2} s} - e^{\sqrt{2} s t} \right].
\end{aligned}
$$

When $t < \frac{\tau}{s + \operatorname{Tr} \boldsymbol{\Sigma}} \ln \frac{1}{u_0}$, we have $\left\| \boldsymbol{W} - \widetilde{\boldsymbol{W}} \right\| < C_1 u_0^2$ for some constant $C_1$.

We are now ready to bound the difference between the exact solution and an exponential function along one direction

$$
\begin{aligned}
&\boldsymbol{W} - e^{\frac{\alpha+1}{2\tau} s t} \begin{bmatrix} \bar{\boldsymbol{\beta}} \\ 1 \end{bmatrix} \boldsymbol{r}_1^\top \\
&= \left( \boldsymbol{W} - \widetilde{\boldsymbol{W}} \right) + \left( \widetilde{\boldsymbol{W}} - e^{-\frac{\alpha+1}{2\tau} s t} \begin{bmatrix} \bar{\boldsymbol{\beta}} \\ 1 \end{bmatrix} \boldsymbol{r}_1^\top \right) \\
&= \left( \boldsymbol{W} - \widetilde{\boldsymbol{W}} \right) + \frac{1}{2} e^{-\frac{\alpha+1}{2\tau} s t} \begin{bmatrix} \bar{\boldsymbol{\beta}} \\ -1 \end{bmatrix} \left( \bar{\boldsymbol{\beta}}^\top \boldsymbol{W}_1^\top(0) - \boldsymbol{W}_2(0) \right) + \begin{bmatrix} \left( \boldsymbol{I} - \bar{\boldsymbol{\beta}} \bar{\boldsymbol{\beta}}^\top \right) \boldsymbol{W}_1^\top(0) \\ 0 \end{bmatrix}.
\end{aligned}
$$

The first term arises from our approximation of dropping the cubic terms in the dynamics. Its norm is bounded by $C_1 w_{\text{init}}^2$. The second term arises from initialization, which is $O(w_{\text{init}})$. Via triangle inequality, the norm of the total error is of order $O(w_{\text{init}})$.

$$
\left\| \boldsymbol{W} - e^{\frac{\alpha+1}{2\tau} s t} \begin{bmatrix} \bar{\boldsymbol{\beta}} \\ 1 \end{bmatrix} \boldsymbol{r}_1^\top \right\| < C_1 w_{\text{init}}^2 + C_2 w_{\text{init}} < C w_{\text{init}}.
$$

$\square$

Lemma 6 implies two messages. Firstly, the (leaky) ReLU network has the same time-course solution as its linear counterpart except a scale factor determined by $\alpha$, which is consistent with Theorem 8. Secondly, the (leaky) ReLU and linear networks form rank-one weights with small errors in the early phase. We exploit the rank-one weights to reduce the learning dynamics to Equation (12).

## C.2 Learning Dynamics: Late Phase

*Proof of Theorem 8 (square loss).* Theorem 8 relies on Condition 3 and Assumption 7 and arrives at three statements: implementing the same function as in Equation (13), having the same weights as in Equation (14), and retaining rank-one weights as Assumption 7. We prove them one by one.

Part 1: We first prove that the (leaky) ReLU network and the linear network implement the same linear function except scaling when their weights satisfy Assumption 7. Denote $\boldsymbol{W}_2 = [\boldsymbol{W}_2^+, \boldsymbol{W}_2^-]$ where $\boldsymbol{W}_2^+$ are

the positive elements in $\boldsymbol{W}_2$ and $\boldsymbol{W}_2^-$ are the negative elements in $\boldsymbol{W}_2$. For a (leaky) ReLU network with rank-one weights satisfying Assumption 7, we have

$$f(\boldsymbol{x}; \boldsymbol{W}) = \boldsymbol{W}_2 \sigma(\boldsymbol{W}_1 \boldsymbol{x}) = \boldsymbol{W}_2 \sigma\left(\boldsymbol{W}_2^\top \boldsymbol{r}^\top \boldsymbol{x}\right).$$

Notate the positive and negative half-space as

$$\mathbb{S}^+ = \left\{\boldsymbol{x} \in \mathbb{R}^D \middle| \boldsymbol{r}^\top \boldsymbol{x} \geq 0\right\}, \quad \mathbb{S}^- = \left\{\boldsymbol{x} \in \mathbb{R}^D \middle| \boldsymbol{r}^\top \boldsymbol{x} < 0\right\}. \tag{33}$$

For $\boldsymbol{x} \in \mathbb{S}^+$, we have

$$f(\boldsymbol{x}; \boldsymbol{W}) = \boldsymbol{r}^\top \boldsymbol{x} \boldsymbol{W}_2 \sigma(\boldsymbol{W}_2^\top) = \boldsymbol{r}^\top \boldsymbol{x} \begin{bmatrix} \boldsymbol{W}_2^+ & \boldsymbol{W}_2^- \end{bmatrix} \begin{bmatrix} \boldsymbol{W}_2^{+\top} \\ \alpha \boldsymbol{W}_2^{-\top} \end{bmatrix} = \boldsymbol{r}^\top \boldsymbol{x} \left(\|\boldsymbol{W}_2^+\|^2 + \alpha\|\boldsymbol{W}_2^-\|^2\right).$$

For $\boldsymbol{x} \in \mathbb{S}^-$, we have

$$f(\boldsymbol{x}; \boldsymbol{W}) = -\boldsymbol{r}^\top \boldsymbol{x} \boldsymbol{W}_2 \sigma(-\boldsymbol{W}_2^\top) = \boldsymbol{r}^\top \boldsymbol{x} \begin{bmatrix} \boldsymbol{W}_2^+ & \boldsymbol{W}_2^- \end{bmatrix} \begin{bmatrix} \alpha \boldsymbol{W}_2^{+\top} \\ \boldsymbol{W}_2^{-\top} \end{bmatrix} = \boldsymbol{r}^\top \boldsymbol{x} \left(\alpha\|\boldsymbol{W}_2^+\|^2 + \|\boldsymbol{W}_2^-\|^2\right).$$

Under Assumption 7, we have $\|\boldsymbol{W}_2^+\| = \|\boldsymbol{W}_2^-\|$. Hence, the (leaky) ReLU network implements

$$\forall \boldsymbol{r}, \quad f(\boldsymbol{x}; \boldsymbol{W}) = \boldsymbol{W}_2 \sigma(\boldsymbol{W}_1 \boldsymbol{x}) = \frac{\alpha + 1}{2} \boldsymbol{r}^\top \boldsymbol{x} \|\boldsymbol{W}_2\|^2. \tag{34}$$

Under Assumption 7, the linear network implements

$$f^{\text{lin}}(\boldsymbol{x}; \boldsymbol{W}) = \boldsymbol{W}_2 \boldsymbol{W}_1 \boldsymbol{x} = \boldsymbol{W}_2 \boldsymbol{W}_2^\top \boldsymbol{r}^\top \boldsymbol{x} = \boldsymbol{r}^\top \boldsymbol{x} \|\boldsymbol{W}_2\|^2. \tag{35}$$

Comparing Equations (34) and (35), we find that when the weights satisfy Assumption 7, the (leaky) ReLU network implements the same function as the linear network except a scale factor

$$\boldsymbol{W}_2 \sigma(\boldsymbol{W}_1 \boldsymbol{x}) = \frac{\alpha + 1}{2} \boldsymbol{W}_2 \boldsymbol{W}_1 \boldsymbol{x}. \tag{36}$$

Part 2: We then look into the learning dynamics to prove that the weights in the (leaky) ReLU and the linear network are the same except scaling. Substituting Equation (36) into the dynamics, we get

$$\begin{aligned} \tau \dot{\boldsymbol{W}}_1 &= \frac{\alpha + 1}{2} \boldsymbol{W}_2^\top \boldsymbol{\beta}^\top - \left\langle \sigma'(\boldsymbol{W}_1 \boldsymbol{x}) \odot \boldsymbol{W}_2^\top \boldsymbol{W}_2 \sigma(\boldsymbol{W}_1 \boldsymbol{x}) \boldsymbol{x}^\top \right\rangle \\ &= \frac{\alpha + 1}{2} \boldsymbol{W}_2^\top \boldsymbol{\beta}^\top - \frac{\alpha + 1}{2} \left\langle \sigma'(\boldsymbol{W}_1 \boldsymbol{x}) \odot \boldsymbol{W}_2^\top \boldsymbol{W}_2 \boldsymbol{W}_1 \boldsymbol{x} \boldsymbol{x}^\top \right\rangle, \end{aligned} \tag{37a}$$

$$\begin{aligned} \tau \dot{\boldsymbol{W}}_2 &= \frac{\alpha + 1}{2} \boldsymbol{\beta}^\top \boldsymbol{W}_1^\top - \boldsymbol{W}_2 \left\langle \sigma(\boldsymbol{W}_1 \boldsymbol{x}) \sigma(\boldsymbol{W}_1 \boldsymbol{x})^\top \right\rangle \\ &= \frac{\alpha + 1}{2} \boldsymbol{\beta}^\top \boldsymbol{W}_1^\top - \frac{\alpha + 1}{2} \boldsymbol{W}_2 \boldsymbol{W}_1 \left\langle \boldsymbol{x} \sigma(\boldsymbol{W}_1 \boldsymbol{x})^\top \right\rangle. \end{aligned} \tag{37b}$$

We compute the second terms in the dynamics under Condition 3 and Assumption 7

$$\begin{aligned} \left\langle \sigma'(\boldsymbol{W}_1 \boldsymbol{x}) \odot \boldsymbol{W}_2^\top \boldsymbol{W}_2 \boldsymbol{W}_1 \boldsymbol{x} \boldsymbol{x}^\top \right\rangle &= \frac{1}{2} \left\langle \sigma'(\boldsymbol{W}_1 \boldsymbol{x}) \odot \boldsymbol{W}_2^\top \boldsymbol{W}_2 \boldsymbol{W}_1 \boldsymbol{x} \boldsymbol{x}^\top \right\rangle_{\mathbb{S}^+} + \frac{1}{2} \left\langle \sigma'(\boldsymbol{W}_1 \boldsymbol{x}) \odot \boldsymbol{W}_2^\top \boldsymbol{W}_2 \boldsymbol{W}_1 \boldsymbol{x} \boldsymbol{x}^\top \right\rangle_{\mathbb{S}^-} \\ &= \frac{1}{2} \begin{bmatrix} \boldsymbol{1} \\ \alpha \boldsymbol{1} \end{bmatrix} \odot \boldsymbol{W}_2^\top \boldsymbol{W}_2 \boldsymbol{W}_1 \left\langle \boldsymbol{x} \boldsymbol{x}^\top \right\rangle_{\mathbb{S}^+} + \frac{1}{2} \begin{bmatrix} \alpha \boldsymbol{1} \\ \boldsymbol{1} \end{bmatrix} \odot \boldsymbol{W}_2^\top \boldsymbol{W}_2 \boldsymbol{W}_1 \left\langle \boldsymbol{x} \boldsymbol{x}^\top \right\rangle_{\mathbb{S}^-} \\ &= \frac{1}{2} \begin{bmatrix} \boldsymbol{W}_2^{+\top} \\ \alpha \boldsymbol{W}_2^{-\top} \end{bmatrix} \boldsymbol{W}_2 \boldsymbol{W}_1 \boldsymbol{\Sigma} + \frac{1}{2} \begin{bmatrix} \alpha \boldsymbol{W}_2^{+\top} \\ \boldsymbol{W}_2^{-\top} \end{bmatrix} \boldsymbol{W}_2 \boldsymbol{W}_1 \boldsymbol{\Sigma} \\ &= \frac{\alpha + 1}{2} \boldsymbol{W}_2^\top \boldsymbol{W}_2 \boldsymbol{W}_1 \boldsymbol{\Sigma}, \end{aligned} \tag{38}$$

and

$$
\begin{aligned}
\left\langle \boldsymbol{x}\sigma(\boldsymbol{W}_1\boldsymbol{x})^\top \right\rangle &= \frac{1}{2}\left\langle \boldsymbol{x}\sigma(\boldsymbol{W}_1\boldsymbol{x})^\top \right\rangle_{\mathbb{S}^+} + \frac{1}{2}\left\langle \boldsymbol{x}\sigma(\boldsymbol{W}_1\boldsymbol{x})^\top \right\rangle_{\mathbb{S}^-} \\
&= \frac{1}{2}\left\langle \boldsymbol{x}\boldsymbol{x}^\top \right\rangle_{\mathbb{S}^+} \boldsymbol{r}\begin{bmatrix} \boldsymbol{W}_2^+ & \alpha\boldsymbol{W}_2^- \end{bmatrix} + \frac{1}{2}\left\langle \boldsymbol{x}\boldsymbol{x}^\top \right\rangle_{\mathbb{S}^-} \boldsymbol{r}\begin{bmatrix} \alpha\boldsymbol{W}_2^+ & \boldsymbol{W}_2^- \end{bmatrix} \\
&= \frac{\alpha+1}{2}\boldsymbol{\Sigma}\boldsymbol{r}\boldsymbol{W}_2 \\
&= \frac{\alpha+1}{2}\boldsymbol{\Sigma}\boldsymbol{W}_1^\top.
\end{aligned}
\tag{39}
$$

Substituting Equations (38) and (39) into Equation (37), we reduce the dynamics to

$$
\tau\dot{\boldsymbol{W}}_1 = \frac{\alpha+1}{2}\boldsymbol{W}_2^\top\boldsymbol{\beta}^\top - \left(\frac{\alpha+1}{2}\right)^2 \boldsymbol{W}_2^\top\boldsymbol{W}_2\boldsymbol{W}_1\boldsymbol{\Sigma},
$$

$$
\tau\dot{\boldsymbol{W}}_2 = \frac{\alpha+1}{2}\boldsymbol{\beta}^\top\boldsymbol{W}_1^\top - \left(\frac{\alpha+1}{2}\right)^2 \boldsymbol{W}_2\boldsymbol{W}_1\boldsymbol{\Sigma}\boldsymbol{W}_1^\top.
$$

This is the same expression as Equation (12) in the main text. If we scale the weights

$$
\overline{\boldsymbol{W}}_1 = \sqrt{\frac{\alpha+1}{2}}\boldsymbol{W}_1, \quad \overline{\boldsymbol{W}}_2 = \sqrt{\frac{\alpha+1}{2}}\boldsymbol{W}_2,
\tag{40}
$$

the scaled (leaky) ReLU network dynamics $\overline{\boldsymbol{W}}(t)$ is the same as that of a linear network given in Equation (3) except for a different time constant

$$
\frac{2\tau}{\alpha+1}\dot{\overline{\boldsymbol{W}}}_1 = \overline{\boldsymbol{W}}_2^\top\left(\boldsymbol{\beta}^\top - \overline{\boldsymbol{W}}_2\overline{\boldsymbol{W}}_1\boldsymbol{\Sigma}\right), \quad \frac{2\tau}{\alpha+1}\dot{\overline{\boldsymbol{W}}}_2 = \left(\boldsymbol{\beta}^\top - \overline{\boldsymbol{W}}_2\overline{\boldsymbol{W}}_1\boldsymbol{\Sigma}\right)\overline{\boldsymbol{W}}_1^\top.
$$

Because Theorem 8 defines the initial condition $\overline{\boldsymbol{W}}(0) = \sqrt{\frac{\alpha+1}{2}}\boldsymbol{W}(0) = \boldsymbol{W}^{\mathrm{lin}}(0)$, the weights in the linear network and the scaled weights in the (leaky) ReLU network start from the same initialization, obey the same dynamics, and consequently stay the same $\forall\, t \geq 0$

$$
\overline{\boldsymbol{W}}(t) = \boldsymbol{W}^{\mathrm{lin}}\left(\frac{\alpha+1}{2}t\right) \quad \Leftrightarrow \quad \boldsymbol{W}(t) = \sqrt{\frac{2}{\alpha+1}}\boldsymbol{W}^{\mathrm{lin}}\left(\frac{\alpha+1}{2}t\right).
$$

This proves Equation (14). Substituting Equation (14) into Equation (36) proves Equation (13)

$$
\begin{aligned}
f(\boldsymbol{x};\boldsymbol{W}(t)) &= \frac{\alpha+1}{2}\boldsymbol{W}(t)\boldsymbol{W}(t)\boldsymbol{x} = \boldsymbol{W}_2^{\mathrm{lin}}\left(\frac{\alpha+1}{2}t\right)\boldsymbol{W}_1^{\mathrm{lin}}\left(\frac{\alpha+1}{2}t\right)\boldsymbol{x} \\
&\equiv f^{\mathrm{lin}}\left(\boldsymbol{x};\boldsymbol{W}^{\mathrm{lin}}\left(\frac{\alpha+1}{2}t\right)\right).
\end{aligned}
$$

Part 3: We show that Assumption 7 made at time $t = 0$ remains valid for $t > 0$, meaning that weights which start with rank-one structure remain rank-one. With Assumption 7 at time $t = 0$, the dynamics of the (leaky) ReLU network is described by Equation (12). This dynamics is the same as scaled dynamics in a linear network and thus satisfies the balancing property of linear networks (Ji & Telgarsky, 2019; Du et al., 2018)

$$
\frac{d}{dt}\left(\boldsymbol{W}_1\boldsymbol{W}_1^\top - \boldsymbol{W}_2^\top\boldsymbol{W}_2\right) = 0.
\tag{41}
$$

Under Assumption 7 at time $t = 0$, this quantity is zero at time $t = 0$ and will stay zero

$$
\forall\, t \geq 0: \quad \boldsymbol{W}_1(t)\boldsymbol{W}_1(t)^\top - \boldsymbol{W}_2(t)^\top\boldsymbol{W}_2(t) = \boldsymbol{W}_1(0)\boldsymbol{W}_1(0)^\top - \boldsymbol{W}_2(0)^\top\boldsymbol{W}_2(0) = \boldsymbol{0}.
$$

Because $\text{rank}(\boldsymbol{W}_1\boldsymbol{W}_1^\top) = \text{rank}(\boldsymbol{W}_1)$, the balancing property enforces that $\boldsymbol{W}_1$ and $\boldsymbol{W}_2$ have equal rank. Since $\boldsymbol{W}_2$ is a vector, $\boldsymbol{W}_1$ must also have rank one. We can write a rank-one matrix as the outer-product of two vectors $\boldsymbol{W}_1 = \boldsymbol{v}\boldsymbol{r}^\top$. We can assume $\|\boldsymbol{r}\| = 1$ for convenience and get

$$\boldsymbol{W}_1\boldsymbol{W}_1^\top = \boldsymbol{v}\boldsymbol{r}^\top\boldsymbol{r}\boldsymbol{v}^\top = \boldsymbol{v}\boldsymbol{v}^\top = \boldsymbol{W}_2^\top\boldsymbol{W}_2 \quad \Rightarrow \quad \boldsymbol{v} = \pm\boldsymbol{W}_2^\top.$$

Because Assumption 7 specifies $\boldsymbol{W}_1 = \boldsymbol{W}_2^\top\boldsymbol{r}^\top$, then $\boldsymbol{v} = \boldsymbol{W}_2^\top$. To summarize, Assumption 7 at time $t = 0$ reduces the learning dynamics of the ReLU network to be similar to that of a linear network. The reduced dynamics satisfies the balancing property which enforces that the weights remain rank-one, thus satisfying Assumption 7 for all $t \geq 0$. $\qquad\square$

*Proof of Theorem 8 (logistic loss).* We prove Theorem 8 for logistic loss $\mathcal{L}_{\text{LG}} = \langle \ln(1 + e^{-y\hat{y}}) \rangle$.

Part 1: Same as the square loss case because proving Equation (36) does not involve the loss function.

Part 2: The gradient flow dynamics of a two-layer linear network trained with logistic loss are

$$\tau\dot{\boldsymbol{W}}_1^{\text{lin}} = \boldsymbol{W}_2^{\text{lin}\top}\boldsymbol{W}_2^{\text{lin}}\boldsymbol{W}_1^{\text{lin}}\left\langle \frac{\boldsymbol{x}\boldsymbol{x}^\top}{e^{y\boldsymbol{W}_2^{\text{lin}}\boldsymbol{W}_1^{\text{lin}}\boldsymbol{x}} + 1} \right\rangle, \tag{42a}$$

$$\tau\dot{\boldsymbol{W}}_2^{\text{lin}} = \boldsymbol{W}_2^{\text{lin}}\boldsymbol{W}_1^{\text{lin}}\left\langle \frac{\boldsymbol{x}\boldsymbol{x}^\top}{e^{y\boldsymbol{W}_2^{\text{lin}}\boldsymbol{W}_1^{\text{lin}}\boldsymbol{x}} + 1} \right\rangle\boldsymbol{W}_1^{\text{lin}\top}. \tag{42b}$$

The gradient flow dynamics of a two-layer (leaky) ReLU network trained with logistic loss are

$$\tau\dot{\boldsymbol{W}}_1 = \left\langle \frac{\sigma'(\boldsymbol{W}_1\boldsymbol{x}) \odot \boldsymbol{W}_2^\top\boldsymbol{W}_2\sigma(\boldsymbol{W}_1\boldsymbol{x})\boldsymbol{x}^\top}{e^{y\boldsymbol{W}_2\sigma(\boldsymbol{W}_1\boldsymbol{x})} + 1} \right\rangle \tag{43a}$$

$$\tau\dot{\boldsymbol{W}}_2 = \boldsymbol{W}_2\left\langle \frac{\sigma(\boldsymbol{W}_1\boldsymbol{x})\sigma(\boldsymbol{W}_1\boldsymbol{x})^\top}{e^{y\boldsymbol{W}_2\sigma(\boldsymbol{W}_1\boldsymbol{x})} + 1} \right\rangle \tag{43b}$$

Substituting Equation (36) into Equation (43), we get

$$\tau\dot{\boldsymbol{W}}_1 = \frac{\alpha+1}{2}\left\langle \frac{\sigma'(\boldsymbol{W}_1\boldsymbol{x}) \odot \boldsymbol{W}_2^\top\boldsymbol{W}_2\boldsymbol{W}_1\boldsymbol{x}\boldsymbol{x}^\top}{e^{\frac{\alpha+1}{2}y\boldsymbol{W}_2\boldsymbol{W}_1\boldsymbol{x}} + 1} \right\rangle \tag{44a}$$

$$\tau\dot{\boldsymbol{W}}_2 = \frac{\alpha+1}{2}\boldsymbol{W}_2\boldsymbol{W}_1\left\langle \frac{\boldsymbol{x}\sigma(\boldsymbol{W}_1\boldsymbol{x})^\top}{e^{\frac{\alpha+1}{2}y\boldsymbol{W}_2\boldsymbol{W}_1\boldsymbol{x}} + 1} \right\rangle \tag{44b}$$

Under Condition 3 and Assumption 7, Equation (44) can be simplified

$$\left\langle \frac{\sigma'(\boldsymbol{W}_1\boldsymbol{x}) \odot \boldsymbol{W}_2^\top\boldsymbol{W}_2\boldsymbol{W}_1\boldsymbol{x}\boldsymbol{x}^\top}{e^{\frac{\alpha+1}{2}y\boldsymbol{W}_2\boldsymbol{W}_1\boldsymbol{x}} + 1} \right\rangle$$

$$= \frac{1}{2}\left\langle \frac{\sigma'(\boldsymbol{W}_1\boldsymbol{x}) \odot \boldsymbol{W}_2^\top\boldsymbol{W}_2\boldsymbol{W}_1\boldsymbol{x}\boldsymbol{x}^\top}{e^{\frac{\alpha+1}{2}y\boldsymbol{W}_2\boldsymbol{W}_1\boldsymbol{x}} + 1} \right\rangle_{\mathbb{S}^+} + \frac{1}{2}\left\langle \frac{\sigma'(\boldsymbol{W}_1\boldsymbol{x}) \odot \boldsymbol{W}_2^\top\boldsymbol{W}_2\boldsymbol{W}_1\boldsymbol{x}\boldsymbol{x}^\top}{e^{\frac{\alpha+1}{2}y\boldsymbol{W}_2\boldsymbol{W}_1\boldsymbol{x}} + 1} \right\rangle_{\mathbb{S}^-}$$

$$= \frac{1}{2}\begin{bmatrix} \alpha\boldsymbol{W}_2^{+\top} \\ \boldsymbol{W}_2^{-\top} \end{bmatrix}\boldsymbol{W}_2\boldsymbol{W}_1\left\langle \frac{\boldsymbol{x}\boldsymbol{x}^\top}{e^{\frac{\alpha+1}{2}y\boldsymbol{W}_2\boldsymbol{W}_1\boldsymbol{x}} + 1} \right\rangle_{\mathbb{S}^+} + \frac{1}{2}\begin{bmatrix} \boldsymbol{W}_2^{+\top} \\ \alpha\boldsymbol{W}_2^{-\top} \end{bmatrix}\boldsymbol{W}_2\boldsymbol{W}_1\left\langle \frac{\boldsymbol{x}\boldsymbol{x}^\top}{e^{\frac{\alpha+1}{2}y\boldsymbol{W}_2\boldsymbol{W}_1\boldsymbol{x}} + 1} \right\rangle_{\mathbb{S}^-}$$

$$= \frac{\alpha+1}{2}\boldsymbol{W}_2^\top\boldsymbol{W}_2\boldsymbol{W}_1\left\langle \frac{\boldsymbol{x}\boldsymbol{x}^\top}{e^{\frac{\alpha+1}{2}y\boldsymbol{W}_2\boldsymbol{W}_1\boldsymbol{x}} + 1} \right\rangle,$$

and

$$\left\langle \frac{\boldsymbol{x}\sigma(\boldsymbol{W}_1\boldsymbol{x})^\top}{e^{\frac{\alpha+1}{2}y\boldsymbol{W}_2\boldsymbol{W}_1\boldsymbol{x}} + 1} \right\rangle$$

$$= \frac{1}{2}\left\langle \frac{\boldsymbol{x}\sigma(\boldsymbol{W}_1\boldsymbol{x})^\top}{e^{\frac{\alpha+1}{2}y\boldsymbol{W}_2\boldsymbol{W}_1\boldsymbol{x}} + 1} \right\rangle_{\mathbb{S}^+} + \frac{1}{2}\left\langle \frac{\boldsymbol{x}\sigma(\boldsymbol{W}_1\boldsymbol{x})^\top}{e^{\frac{\alpha+1}{2}y\boldsymbol{W}_2\boldsymbol{W}_1\boldsymbol{x}} + 1} \right\rangle_{\mathbb{S}^-}$$

$$= \frac{1}{2}\left\langle \frac{\boldsymbol{x}\boldsymbol{x}^\top}{e^{\frac{\alpha+1}{2}y\boldsymbol{W}_2\boldsymbol{W}_1\boldsymbol{x}} + 1} \right\rangle_{\mathbb{S}^+}\boldsymbol{r}\begin{bmatrix} \alpha\boldsymbol{W}_2^+ & \boldsymbol{W}_2^- \end{bmatrix} + \frac{1}{2}\left\langle \frac{\boldsymbol{x}\boldsymbol{x}^\top}{e^{\frac{\alpha+1}{2}y\boldsymbol{W}_2\boldsymbol{W}_1\boldsymbol{x}} + 1} \right\rangle_{\mathbb{S}^-}\boldsymbol{r}\begin{bmatrix} \boldsymbol{W}_2^+ & \alpha\boldsymbol{W}_2^- \end{bmatrix}$$

$$= \frac{\alpha+1}{2}\left\langle \frac{\boldsymbol{x}\boldsymbol{x}^\top}{e^{\frac{\alpha+1}{2}y\boldsymbol{W}_2\boldsymbol{W}_1\boldsymbol{x}} + 1} \right\rangle\boldsymbol{W}_1^\top.$$

Thus, the reduced dynamics of the two-layer (leaky) ReLU network are

$$\tau\dot{\boldsymbol{W}}_1 = \left(\frac{\alpha+1}{2}\right)^2 \boldsymbol{W}_2^\top \boldsymbol{W}_2 \boldsymbol{W}_1 \left\langle \frac{\boldsymbol{x}\boldsymbol{x}^\top}{e^{\frac{\alpha+1}{2}y\boldsymbol{W}_2\boldsymbol{W}_1\boldsymbol{x}}+1} \right\rangle,$$

$$\tau\dot{\boldsymbol{W}}_2 = \left(\frac{\alpha+1}{2}\right)^2 \boldsymbol{W}_2 \boldsymbol{W}_1 \left\langle \frac{\boldsymbol{x}\boldsymbol{x}^\top}{e^{\frac{\alpha+1}{2}y\boldsymbol{W}_2\boldsymbol{W}_1\boldsymbol{x}}+1} \right\rangle \boldsymbol{W}_1^\top.$$

If we scale the weights as Equation (40), the (leaky) ReLU network dynamics is the same as that of a linear network given in Equation (42) except for a different time constant

$$\frac{2\tau}{\alpha+1}\dot{\overline{\boldsymbol{W}}}_1 = \overline{\boldsymbol{W}}_2^\top \overline{\boldsymbol{W}}_2 \overline{\boldsymbol{W}}_1 \left\langle \frac{\boldsymbol{x}\boldsymbol{x}^\top}{e^{y\overline{\boldsymbol{W}}_2\overline{\boldsymbol{W}}_1\boldsymbol{x}}+1} \right\rangle,$$

$$\frac{2\tau}{\alpha+1}\dot{\overline{\boldsymbol{W}}}_2 = \overline{\boldsymbol{W}}_2 \overline{\boldsymbol{W}}_1 \left\langle \frac{\boldsymbol{x}\boldsymbol{x}^\top}{e^{y\overline{\boldsymbol{W}}_2\overline{\boldsymbol{W}}_1\boldsymbol{x}}+1} \right\rangle \overline{\boldsymbol{W}}_1^\top.$$

Through the same reasoning as the square loss case, Equations (13) and (14) are proven.

Part 3: Same as the square loss case. $\qquad\square$

## C.3 Global Minimum

*Proof of Corollary 10.* The converged solution $\boldsymbol{w}^*$ is a direct consequence of the equivalence we showed in Theorem 8 and prior results of linear networks (Saxe et al., 2014; Soudry et al., 2018).

We now show that for symmetric datasets satisfying Condition 3, the global minimum solution of a two-layer bias-free (leaky) ReLU network trained with square loss is linear.

Based on Theorem 1, we can write a two-layer bias-free (leaky) ReLU network as a linear function plus an even function $f(\boldsymbol{x}) = \boldsymbol{x}^\top \boldsymbol{w}^* + f_e(\boldsymbol{x})$ where $f_e(\cdot)$ denotes an even function. For datasets satisfying Condition 3, the square loss is

$$\mathcal{L} = \frac{1}{2}\left\langle \left(y-\boldsymbol{x}^\top\boldsymbol{w}^*-f_e(\boldsymbol{x})\right)^2 \right\rangle_{p(\boldsymbol{x})}$$

$$= \frac{1}{2}\left\langle \left(y-\boldsymbol{x}^\top\boldsymbol{w}^*\right)^2 - 2(y-\boldsymbol{A}\boldsymbol{x})f_e(\boldsymbol{x}) + f_e(\boldsymbol{x})^2 \right\rangle_{p(\boldsymbol{x})}$$

$$= \frac{1}{2}\left\langle \left(y-\boldsymbol{x}^\top\boldsymbol{w}^*\right)^2 \right\rangle_{p(\boldsymbol{x})} + \frac{1}{2}\left\langle f_e(\boldsymbol{x})^2 \right\rangle_{p(\boldsymbol{x})}.$$

The square loss attains its minimum when both $\left\langle \left(y-\boldsymbol{x}^\top\boldsymbol{w}^*\right)^2 \right\rangle_{p(\boldsymbol{x})}$ and $\left\langle f_e(\boldsymbol{x})^2 \right\rangle_{p(\boldsymbol{x})}$ are minimized. The former is minimized when $\boldsymbol{w}^* = \boldsymbol{\Sigma}^{-1}\boldsymbol{\beta}$. The latter is minimized when $f_e(\boldsymbol{x}) = 0$. Hence, for symmetric datasets satisfying Condition 3, the two-layer bias-free (leaky) ReLU network achieves globally minimum square loss with the linear, ordinary least squares solution $f(\boldsymbol{x}) = \boldsymbol{x}^\top\boldsymbol{w}^* = \boldsymbol{x}^\top\boldsymbol{\Sigma}^{-1}\boldsymbol{\beta}$. $\qquad\square$

## C.4 Effect of Regularization

Theorem 8 still holds when L2 regularization is applied. Specifically, Theorem 8 holds if L2 regularization is added with hyperparameter $\lambda_\alpha = \frac{\alpha+1}{2}\lambda$, i.e., the loss is $\mathcal{L}_{\text{reg}} = \mathcal{L} + \frac{\alpha+1}{2}\lambda\|\boldsymbol{W}\|_2^2$. With similar calculations, we find that the regularized dynamics of the two-layer bias-free (leaky) ReLU network is

$$\tau\dot{\boldsymbol{W}}_1 = \frac{\alpha+1}{2}\boldsymbol{W}_2^\top\boldsymbol{\beta}^\top - \left(\frac{\alpha+1}{2}\right)^2 \boldsymbol{W}_2^\top\boldsymbol{W}_2\boldsymbol{W}_1\boldsymbol{\Sigma} - \frac{\alpha+1}{2}\lambda\boldsymbol{W}_1, \tag{45a}$$

$$\tau\dot{\boldsymbol{W}}_2 = \frac{\alpha+1}{2}\boldsymbol{\beta}^\top\boldsymbol{W}_1^\top - \left(\frac{\alpha+1}{2}\right)^2 \boldsymbol{W}_2\boldsymbol{W}_1\boldsymbol{\Sigma}\boldsymbol{W}_1^\top - \frac{\alpha+1}{2}\lambda\boldsymbol{W}_2. \tag{45b}$$

If we scale the weights as Equation (40), the regularized (leaky) ReLU network dynamics is again the same as that of a regularized linear network except for a different time constant

$$\frac{2\tau}{\alpha+1}\dot{\overline{W}}_1 = \overline{W}_2^\top\left(\beta^\top - \overline{W}_2\overline{W}_1\Sigma\right) - \lambda\overline{W}_1, \quad \frac{2\tau}{\alpha+1}\dot{\overline{W}}_2 = \left(\beta^\top - \overline{W}_2\overline{W}_1\Sigma\right)\overline{W}_1^\top - \lambda\overline{W}_2.$$

We validate this with simulations in Figure 7a. As in the unregularized case, we find that the loss curves with different leaky ReLU slopes collapse to one curve after rescaling time and the differences between weight matrices are small.

We note a concurrent work (Wang et al., 2025) that considers two-layer bias-free ReLU networks with L2 regularization.

## C.5 Effect of Learning Rate

We empirically find that with a moderately large learning rate, the behaviors of two-layer bias-free (leaky) ReLU networks are consistent with Theorem 8. For simulations in Figure 7b, we use a learning rate of 0.6, which is 150 times larger than the learning rate used in Figure 3, 0.004. Due to the larger learning rate, the loss curves in Figure 7b is less smooth than those in Figures 3, 7a and 7c. Nonetheless, the loss curves with different leaky ReLU slopes collapse to one curve after rescaling time and the differences between weight matrices are small.

If the learning rate is further increased, oscillations in the loss curves occur, suggesting unstable training. In such cases, the equivalence described in Theorem 8 no longer holds. However, the learning rate is typically chosen to avoid such oscillations in training.

## C.6 Effect of Initialization

We empirically find that under large initialization, two-layer bias-free (leaky) ReLU networks still have similar learning dynamics as its linear counterpart. As in the small initialization case, the loss curves in Figure 7c with different leaky ReLU slopes collapse to one curve after rescaling time. The differences between weight matrices are larger than those in the case of small initialization but are still less than 3%. With large initialization and a moderately large learning rate, the behavior remains consistent, as shown in Figure 7d.

This is related to the limited expressivity of bias-free ReLU networks. Within the expressivity of two-layer bias-free ReLU networks, the linear solution is the global minimum for symmetric datasets. The two-layer bias-free ReLU network learns the linear solution starting from either small or large initialization.

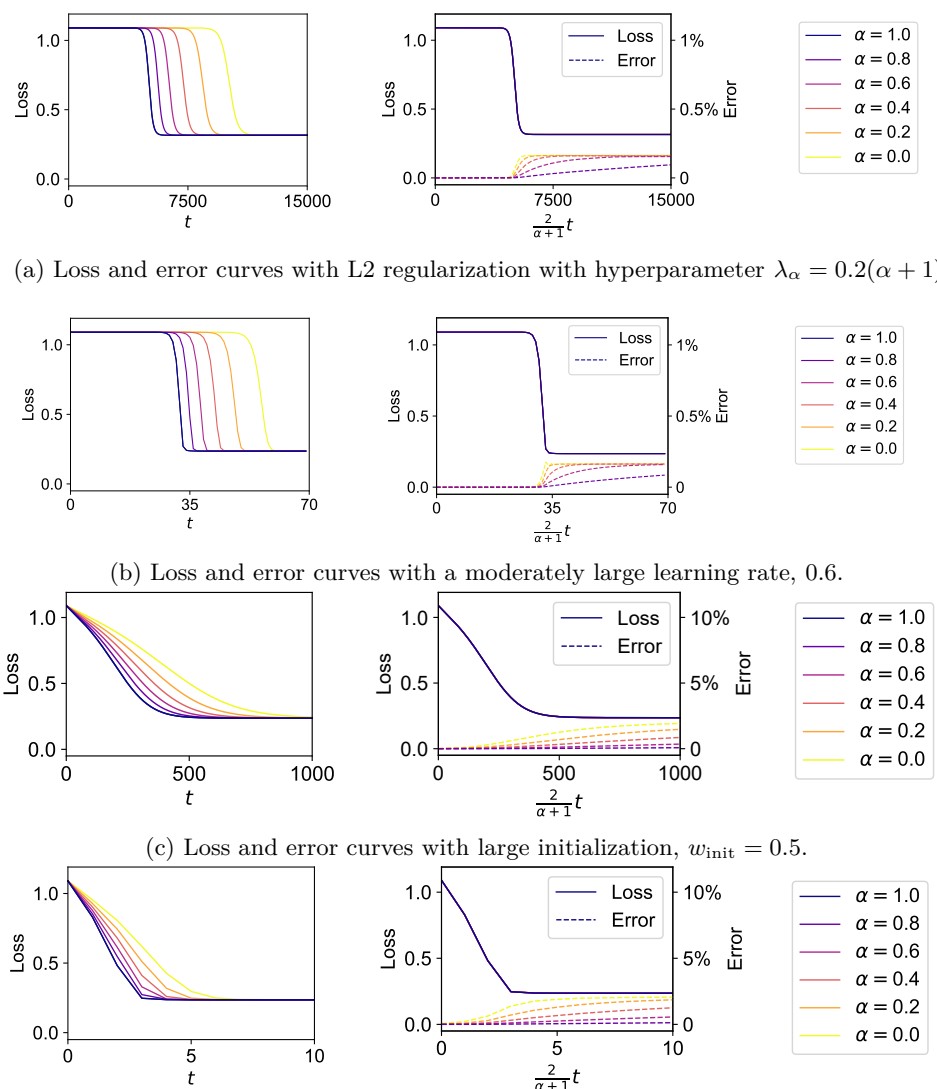

(a) Loss and error curves with L2 regularization with hyperparameter $\lambda_\alpha = 0.2(\alpha + 1)$.

(b) Loss and error curves with a moderately large learning rate, 0.6.

(c) Loss and error curves with large initialization, $w_{\text{init}} = 0.5$.

(d) Loss and error curves with large initialization, $w_{\text{init}} = 0.5$, and a moderately large learning rate, 0.6.

Figure 7: Two-layer bias-free (leaky) ReLU networks evolve like a linear network even when some of the assumptions in Theorem 8 are lifted. The setup is the same as Figure 3 except for the condition(s) specified in each individual subcaption. In (b,c,d), the time rescaling is implemented by inversely rescaling the learning rate. This avoids the inaccuracy induced by rounding the rescaled time to an integer number of epoch, which becomes non-negligible in the case of a small total number of epochs. In (c,d), with large initialization, the errors between weight matrices are larger than those in the case of small initialization but are still less than 3%.

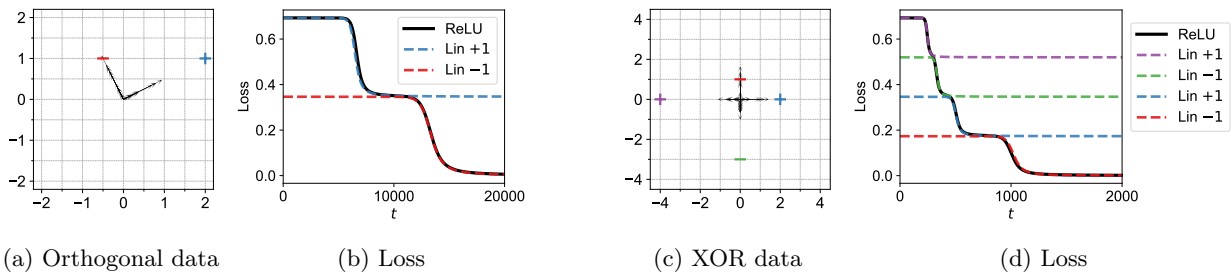

(a) Orthogonal data      (b) Loss      (c) XOR data      (d) Loss

Figure 8: The same as Figure 4 but with logistic loss.

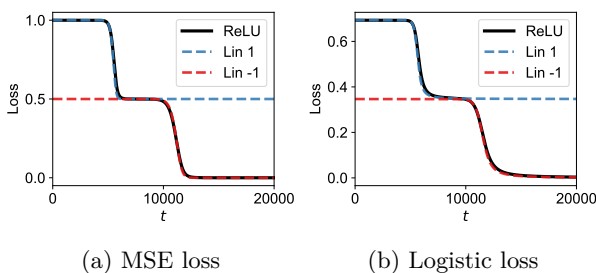

(a) MSE loss      (b) Logistic loss

Figure 9: Two-layer bias-free ReLU networks evolve like two linear networks when trained on a dataset with pairwise orthogonal input, that is $\forall \mu \neq \nu, \boldsymbol{x}_\mu^\top \boldsymbol{x}_\nu = 0$. The dataset has 20 samples, with 10 samples labeled $+1$ and 10 samples labeled $-1$. The input points with $+1$ label have a larger L2 norm than those with $-1$ label. The black curve is the loss of a two-layer bias-free ReLU network trained on all 20 samples. The blue dashed curve is the loss of a two-layer linear network trained on the ten data points with $+1$ label; and the red dashed curve is that trained on the ten data points with $-1$ label. The loss curve of the ReLU network overlaps with those of the two linear networks. (a) Mean square error loss trajectory. (b) Logistic loss trajectory.

## D    Two-Layer Networks Learning Dynamics on Orthogonal Datasets

**Assumption 17.** A two-layer bias-free ReLU network is trained from small initialization on a dataset with orthonormal input, that is $\boldsymbol{x}_\mu^\top \boldsymbol{x}_\nu = 0$ if $\mu \neq \nu$, and $\boldsymbol{x}_\mu^\top \boldsymbol{x}_\mu = 1$. We assume that the weights at time $t_0$ during training have the following form:

$$\boldsymbol{W}_1 = \begin{bmatrix} \boldsymbol{W}_1^A \\ \boldsymbol{W}_1^B \end{bmatrix} = \begin{bmatrix} u_A \boldsymbol{r}_A \bar{\boldsymbol{\beta}}_A^\top \\ u_B \boldsymbol{r}_B \bar{\boldsymbol{\beta}}_B^\top \end{bmatrix}, \quad \boldsymbol{W}_2 = \begin{bmatrix} \boldsymbol{W}_2^A & \boldsymbol{W}_2^B \end{bmatrix} = \begin{bmatrix} u_A \boldsymbol{r}_A^\top & -u_B \boldsymbol{r}_B^\top \end{bmatrix}, \tag{46}$$

where $u > 0$, $\|\boldsymbol{r}_A\| = \|\boldsymbol{r}_B\| = 1$ and all entries in $\boldsymbol{r}_A, \boldsymbol{r}_B$ are non-negative. The vectors $\boldsymbol{\beta}_A, \boldsymbol{\beta}_B$ are defined as

$$\bar{\boldsymbol{\beta}}_A = \frac{\sum_{\mu \in \mathcal{S}_A} y_\mu \boldsymbol{x}_\mu}{\left\| \sum_{\mu \in \mathcal{S}_A} y_\mu \boldsymbol{x}_\mu \right\|}, \quad \mathcal{S}_A = \{\mu | y_\mu > 0\}, \tag{47a}$$

$$\bar{\boldsymbol{\beta}}_B = -\frac{\sum_{\mu \in \mathcal{S}_B} y_\mu \boldsymbol{x}_\mu}{\left\| \sum_{\mu \in \mathcal{S}_B} y_\mu \boldsymbol{x}_\mu \right\|}, \quad \mathcal{S}_B = \{\mu | y_\mu < 0\}. \tag{47b}$$

Here the set $\mathcal{S}_A$ denotes the indices of the data points with positive target output, and $\mathcal{S}_B$ denotes those with negative target output.

**Proposition 18.** *Under Assumption 17, for all $t \geq t_0$, the weights of the two-layer bias-free ReLU network will maintain the form in Equation* (46) *and its learning dynamics is described by*

$$\tau \dot{\boldsymbol{W}}_1^C = \boldsymbol{W}_2^{C\top} \left( \boldsymbol{\beta}_C^\top - \boldsymbol{W}_2^C \boldsymbol{W}_1^C \boldsymbol{\Sigma}_C \right), \quad \tau \dot{\boldsymbol{W}}_2^C = \left( \boldsymbol{\beta}_C^\top - \boldsymbol{W}_2^C \boldsymbol{W}_1^C \boldsymbol{\Sigma}_C \right) \boldsymbol{W}_1^{C\top}, \tag{48}$$

*where* C *represents either* A *or* B*, and*

$$\boldsymbol{\beta}_{\mathrm{C}} = \frac{1}{P} \sum_{\mu \in \mathcal{S}_{\mathrm{C}}} y_\mu \boldsymbol{x}_\mu, \quad \boldsymbol{\Sigma}_{\mathrm{C}} = \frac{1}{P} \sum_{\mu \in \mathcal{S}_{\mathrm{C}}} \boldsymbol{x}_\mu \boldsymbol{x}_\mu^\top. \tag{49}$$

*The dynamics of* $\boldsymbol{W}_1^{\mathrm{C}}, \boldsymbol{W}_2^{\mathrm{C}}$ *is the same as that of a two-layer linear network trained with data points in* $\mathcal{S}_{\mathrm{C}}$, *as given by Equation* (3).

*Proof.* Because the weights satisfy Equation (46) and the data points are orthonormal, we have

$$\forall \mu \in \mathcal{S}_{\mathrm{A}}, \quad \sigma\left(\boldsymbol{W}_1^{\mathrm{A}} \boldsymbol{x}_\mu\right) = \boldsymbol{W}_1^{\mathrm{A}} \boldsymbol{x}_\mu, \sigma\left(\boldsymbol{W}_1^{\mathrm{B}} \boldsymbol{x}_\mu\right) = \boldsymbol{0};$$
$$\forall \mu \in \mathcal{S}_{\mathrm{B}}, \quad \sigma\left(\boldsymbol{W}_1^{\mathrm{B}} \boldsymbol{x}_\mu\right) = \boldsymbol{W}_1^{\mathrm{B}} \boldsymbol{x}_\mu, \sigma\left(\boldsymbol{W}_1^{\mathrm{A}} \boldsymbol{x}_\mu\right) = \boldsymbol{0}.$$

For data points in the dataset, the network computes

$$\forall \mu \in \mathcal{S}_{\mathrm{A}}, \quad f(\boldsymbol{x}_\mu; \boldsymbol{W}) \equiv \boldsymbol{W}_2 \sigma(\boldsymbol{W}_1 \boldsymbol{x}_\mu) = \boldsymbol{W}_2^{\mathrm{A}} \sigma\left(\boldsymbol{W}_1^{\mathrm{A}} \boldsymbol{x}_\mu\right) + \boldsymbol{W}_2^{\mathrm{B}} \sigma\left(\boldsymbol{W}_1^{\mathrm{B}} \boldsymbol{x}_\mu\right) = \boldsymbol{W}_2^{\mathrm{A}} \boldsymbol{W}_1^{\mathrm{A}} \boldsymbol{x}_\mu,$$
$$\forall \mu \in \mathcal{S}_{\mathrm{B}}, \quad f(\boldsymbol{x}_\mu; \boldsymbol{W}) \equiv \boldsymbol{W}_2 \sigma(\boldsymbol{W}_1 \boldsymbol{x}_\mu) = \boldsymbol{W}_2^{\mathrm{A}} \sigma\left(\boldsymbol{W}_1^{\mathrm{A}} \boldsymbol{x}_\mu\right) + \boldsymbol{W}_2^{\mathrm{B}} \sigma\left(\boldsymbol{W}_1^{\mathrm{B}} \boldsymbol{x}_\mu\right) = \boldsymbol{W}_2^{\mathrm{B}} \boldsymbol{W}_1^{\mathrm{B}} \boldsymbol{x}_\mu.$$

The learning dynamics of $\boldsymbol{W}_1^{\mathrm{A}}, \boldsymbol{W}_2^{\mathrm{A}}$ can be calculated as

$$\tau \dot{\boldsymbol{W}}_1^{\mathrm{A}} = \frac{1}{P} \sum_{\mu=1}^{P} \sigma'\left(\boldsymbol{W}_1^{\mathrm{A}} \boldsymbol{x}_\mu\right) \odot \boldsymbol{W}_2^{\mathrm{A}^\top} \left(y_\mu - \boldsymbol{W}_2 \sigma(\boldsymbol{W}_1 \boldsymbol{x}_\mu)\right) \boldsymbol{x}_\mu^\top$$

$$= \frac{1}{P} \sum_{\mu \in \mathcal{S}_{\mathrm{A}}} \boldsymbol{W}_2^{\mathrm{A}^\top} \left(y_\mu - \boldsymbol{W}_2^{\mathrm{A}} \boldsymbol{W}_1^{\mathrm{A}} \boldsymbol{x}_\mu\right) \boldsymbol{x}_\mu^\top$$

$$= \boldsymbol{W}_2^{\mathrm{A}^\top} \left(\boldsymbol{\beta}_{\mathrm{A}}^\top - \boldsymbol{W}_2^{\mathrm{A}} \boldsymbol{W}_1^{\mathrm{A}} \boldsymbol{\Sigma}_{\mathrm{A}}\right),$$

$$\tau \dot{\boldsymbol{W}}_2^{\mathrm{A}} = \sum_{\mu=1}^{P} \left(y_\mu - \boldsymbol{W}_2 \sigma(\boldsymbol{W}_1 \boldsymbol{x}_\mu)\right) \sigma\left(\boldsymbol{W}_1^{\mathrm{A}} \boldsymbol{x}_\mu\right)^\top$$

$$= \sum_{\mu \in \mathcal{S}_{\mathrm{A}}} \left(y_\mu - \boldsymbol{W}_2 \sigma(\boldsymbol{W}_1 \boldsymbol{x}_\mu)\right) \sigma\left(\boldsymbol{W}_1^{\mathrm{A}} \boldsymbol{x}_\mu\right)^\top$$

$$= \left(\boldsymbol{\beta}_{\mathrm{A}}^\top - \boldsymbol{W}_2^{\mathrm{A}} \boldsymbol{W}_1^{\mathrm{A}} \boldsymbol{\Sigma}_{\mathrm{A}}\right) \boldsymbol{W}_1^{\mathrm{A}^\top}.$$

This proves Equation (48). We now prove that the weights will maintain the form in Equation (46) for $t \geq t_0$. We first calculate a term that we will need:

$$\bar{\boldsymbol{\beta}}_{\mathrm{A}}^\top \boldsymbol{\Sigma}_{\mathrm{A}} = \frac{\sum_{\mu \in \mathcal{S}_{\mathrm{A}}} y_\mu \boldsymbol{x}_\mu^\top}{\left\| \sum_{\mu \in \mathcal{S}_{\mathrm{A}}} y_\mu \boldsymbol{x}_\mu \right\|} \frac{\sum_{\mu' \in \mathcal{S}_{\mathrm{A}}} \boldsymbol{x}_{\mu'} \boldsymbol{x}_{\mu'}^\top}{P}$$

$$= \frac{\sum_{\mu, \mu' \in \mathcal{S}_{\mathrm{A}}} y_\mu \boldsymbol{x}_\mu^\top \boldsymbol{x}_{\mu'} \boldsymbol{x}_{\mu'}^\top}{\left\| \sum_{\mu \in \mathcal{S}_{\mathrm{A}}} y_\mu \boldsymbol{x}_\mu \right\| P}$$

$$= \frac{\sum_{\mu \in \mathcal{S}_{\mathrm{A}}} y_\mu \boldsymbol{x}_\mu^\top}{\left\| \sum_{\mu \in \mathcal{S}_{\mathrm{A}}} y_\mu \boldsymbol{x}_\mu \right\| P}$$

$$= \frac{1}{P} \bar{\boldsymbol{\beta}}_{\mathrm{A}}^\top. \tag{50}$$

By substituting Equation (46) into Equation (48) and using Equation (50), we obtain

$$\tau \dot{\boldsymbol{W}}_1^{\mathrm{A}} = \tau \frac{d}{dt} u_{\mathrm{A}} \boldsymbol{r}_{\mathrm{A}} \bar{\boldsymbol{\beta}}_{\mathrm{A}}^\top = u_{\mathrm{A}} \boldsymbol{r}_{\mathrm{A}} \left(\boldsymbol{\beta}_{\mathrm{A}}^\top - u_{\mathrm{A}} \boldsymbol{r}_{\mathrm{A}}^\top u_{\mathrm{A}} \boldsymbol{r}_{\mathrm{A}} \bar{\boldsymbol{\beta}}_{\mathrm{A}}^\top \boldsymbol{\Sigma}_{\mathrm{A}}\right) = u_{\mathrm{A}} \boldsymbol{r}_{\mathrm{A}} \bar{\boldsymbol{\beta}}_{\mathrm{A}}^\top \left(\|\boldsymbol{\beta}_{\mathrm{A}}\| - \frac{1}{P} u_{\mathrm{A}}^2\right), \tag{51a}$$

$$\tau \dot{\boldsymbol{W}}_2^{\mathrm{A}} = \tau \frac{d}{dt} u_{\mathrm{A}} \boldsymbol{r}_{\mathrm{A}}^\top = \left(\boldsymbol{\beta}_{\mathrm{A}}^\top - u_{\mathrm{A}} \boldsymbol{r}_{\mathrm{A}}^\top u_{\mathrm{A}} \boldsymbol{r}_{\mathrm{A}} \bar{\boldsymbol{\beta}}_{\mathrm{A}}^\top \boldsymbol{\Sigma}_{\mathrm{A}}\right) u_{\mathrm{A}} \bar{\boldsymbol{\beta}}_{\mathrm{A}} \boldsymbol{r}_{\mathrm{A}}^\top = u_{\mathrm{A}} \left(\|\boldsymbol{\beta}_{\mathrm{A}}\| - \frac{1}{P} u_{\mathrm{A}}^2\right) \boldsymbol{r}_{\mathrm{A}}^\top. \tag{51b}$$

Equating the coefficients in Equation (51) yields a one-dimensional ordinary differential equation about $u_A$:

$$\tau \dot{u}_A = u_A \left( \|\boldsymbol{\beta}_A\| - \frac{1}{P} u_A^2 \right). \tag{52}$$

Hence, for the $\boldsymbol{W}_1^A, \boldsymbol{W}_2^A$ blocks in Equation (46), only the scalar variable $u_A$ evolves while the vectors $\boldsymbol{r}_A, \bar{\boldsymbol{\beta}}_A$ stay unchanged.

With the same approach, we can calculate the learning dynamics of $\boldsymbol{W}_1^B, \boldsymbol{W}_2^B$

$$\tau \dot{\boldsymbol{W}}_1^B = \boldsymbol{W}_2^{B\top} \left( \boldsymbol{\beta}_B^\top - \boldsymbol{W}_2^B \boldsymbol{W}_1^B \boldsymbol{\Sigma}_B \right),$$
$$\tau \dot{\boldsymbol{W}}_2^B = \left( \boldsymbol{\beta}_B^\top - \boldsymbol{W}_2^B \boldsymbol{W}_1^B \boldsymbol{\Sigma}_B \right) \boldsymbol{W}_1^{B\top}.$$

Substituting in Equation (46), we obtain

$$\tau \dot{\boldsymbol{W}}_1^B = \tau \frac{d}{dt} u_B \boldsymbol{r}_B \bar{\boldsymbol{\beta}}_B^\top = -u_B \boldsymbol{r}_B \left( \boldsymbol{\beta}_B^\top + u_B \boldsymbol{r}_B^\top u_B \boldsymbol{r}_B \bar{\boldsymbol{\beta}}_B^\top \boldsymbol{\Sigma}_B \right) = u_B \boldsymbol{r}_B \bar{\boldsymbol{\beta}}_B^\top \left( \|\boldsymbol{\beta}_B\| - \frac{1}{P} u_B^2 \right), \tag{53a}$$

$$\tau \dot{\boldsymbol{W}}_2^B = -\tau \frac{d}{dt} u_B \boldsymbol{r}_B^\top = \left( \boldsymbol{\beta}_B^\top + u_B \boldsymbol{r}_B^\top u_B \boldsymbol{r}_B \bar{\boldsymbol{\beta}}_B^\top \boldsymbol{\Sigma}_B \right) u_B \bar{\boldsymbol{\beta}}_B \boldsymbol{r}_B^\top = u_B \left( -\|\boldsymbol{\beta}_B\| + \frac{1}{P} u_B^2 \right) \boldsymbol{r}_B^\top. \tag{53b}$$

Equating the coefficients in Equation (53) yields a one-dimensional ordinary differential equation about $u_B$:

$$\tau \dot{u}_B = u_B \left( \|\boldsymbol{\beta}_B\| - \frac{1}{P} u_B^2 \right). \tag{54}$$

Hence, for the $\boldsymbol{W}_1^B, \boldsymbol{W}_2^B$ blocks in Equation (46), only the scalar variable $u_B$ evolves while the vectors $\boldsymbol{r}_B, \bar{\boldsymbol{\beta}}_B$ stay unchanged. Consequently, if the weights satisfy Equation (46) at time $t_0$, Equation (46) will hold for all $t \geq t_0$. $\qquad \square$

# E   Deep ReLU Network Learning Dynamics

*Proof of Proposition 12.* According to Equation (18), all weight elements in the intermediate layers $(\boldsymbol{W}_{L-1}, \cdots, \boldsymbol{W}_3, \boldsymbol{W}_2)$ are non-negative numbers. According to the definition of the ReLU activation function, $\sigma(\boldsymbol{W}_1 \boldsymbol{x})$ yields a vector with non-negative numbers. Thus $\boldsymbol{W}_2 \sigma(\boldsymbol{W}_1 \boldsymbol{x})$ yields a vector with non-negative numbers and we have $\sigma(\boldsymbol{W}_2 \sigma(\boldsymbol{W}_1 \boldsymbol{x})) = \boldsymbol{W}_2 \sigma(\boldsymbol{W}_1 \boldsymbol{x})$. Similarly, all subsequent ReLU activation functions can be ignored[2]. With weights satisfying Equation (18), a deep bias-free ReLU network implements

$$f(\boldsymbol{x}; \boldsymbol{W}) \equiv \boldsymbol{W}_L \sigma(\cdots \sigma(\boldsymbol{W}_2 \sigma(\boldsymbol{W}_1 \boldsymbol{x}))) = \boldsymbol{W}_L \cdots \boldsymbol{W}_2 \sigma(\boldsymbol{W}_1 \boldsymbol{x}).$$

We stick to the notation for the positive and negative half-space defined in Equation (33). For $\boldsymbol{x} \in \mathbb{S}^+$, we have

$$f(\boldsymbol{x}; \boldsymbol{W}) = u \boldsymbol{W}_L \cdots \boldsymbol{W}_2 \begin{bmatrix} \boldsymbol{r}_1^+ \\ \boldsymbol{0} \end{bmatrix} \boldsymbol{r}^\top \boldsymbol{x} = \frac{u^{L-1}}{(\sqrt{2})^{L-2}} \boldsymbol{W}_L \begin{bmatrix} \boldsymbol{r}_{L-1}^+ \\ \boldsymbol{0} \end{bmatrix} \boldsymbol{r}^\top \boldsymbol{x} = \left( \frac{u}{\sqrt{2}} \right)^L \boldsymbol{r}^\top \boldsymbol{x}.$$

For $\boldsymbol{x} \in \mathbb{S}^-$, we have

$$f(\boldsymbol{x}; \boldsymbol{W}) = u \boldsymbol{W}_L \cdots \boldsymbol{W}_2 \begin{bmatrix} \boldsymbol{0} \\ \boldsymbol{r}_1^- \end{bmatrix} \boldsymbol{r}^\top \boldsymbol{x} = \frac{u^{L-1}}{(\sqrt{2})^{L-2}} \boldsymbol{W}_L \begin{bmatrix} \boldsymbol{0} \\ \boldsymbol{r}_{L-1}^- \end{bmatrix} \boldsymbol{r}^\top \boldsymbol{x} = \left( \frac{u}{\sqrt{2}} \right)^L \boldsymbol{r}^\top \boldsymbol{x}.$$

Hence, the deep bias-free ReLU network implements a linear function $f(\boldsymbol{x}; \boldsymbol{W}) = \left( \frac{u}{\sqrt{2}} \right)^L \boldsymbol{r}^\top \boldsymbol{x}$. Notice that a deep linear network with such weights implement

$$\boldsymbol{W}_L \cdots \boldsymbol{W}_2 \boldsymbol{W}_1 \boldsymbol{x} = u \boldsymbol{W}_L \cdots \boldsymbol{W}_2 \begin{bmatrix} \boldsymbol{r}_1^+ \\ \boldsymbol{r}_1^- \end{bmatrix} \boldsymbol{r}^\top \boldsymbol{x} = \cdots = \frac{u^{L-1}}{(\sqrt{2})^{L-2}} \boldsymbol{W}_L \boldsymbol{r}_{L-1} \boldsymbol{r}^\top \boldsymbol{x} = \frac{u^L}{(\sqrt{2})^{L-2}} \boldsymbol{r}^\top \boldsymbol{x}.$$

---

[2]The activation functions can be ignored when calculating the network output but cannot be ignored when calculating the gradients. This is because for a ReLU function $\sigma(z) = \max(z, 0)$ and a linear function $\phi(z) = z$, the function values are equal at zero $\sigma(0) = \phi(0)$ but the derivatives are not equal at zero $\sigma'(0) \neq \phi'(0)$.

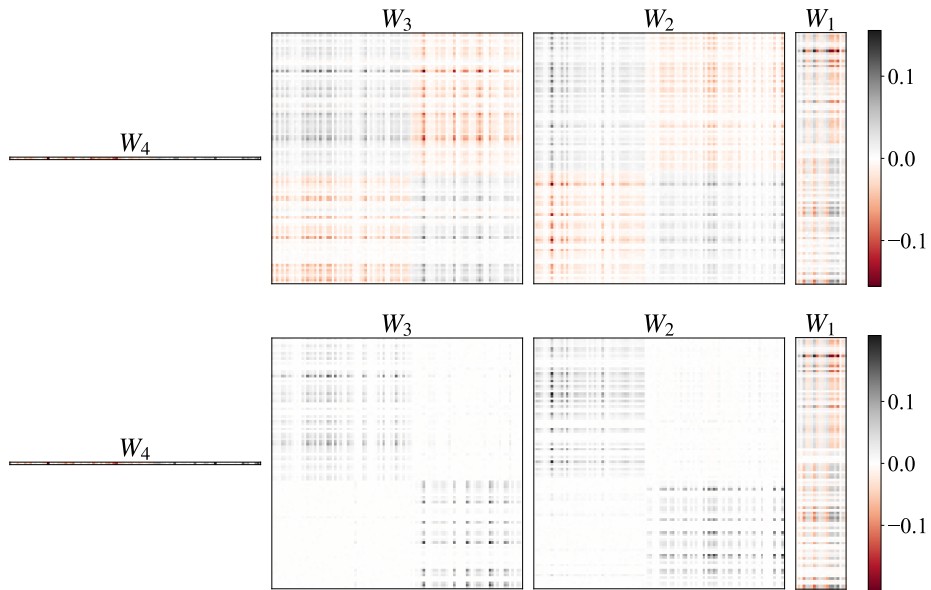

(a) Weights in a 4-layer linear (top) and ReLU (bottom) network as Equations (17) and (18).

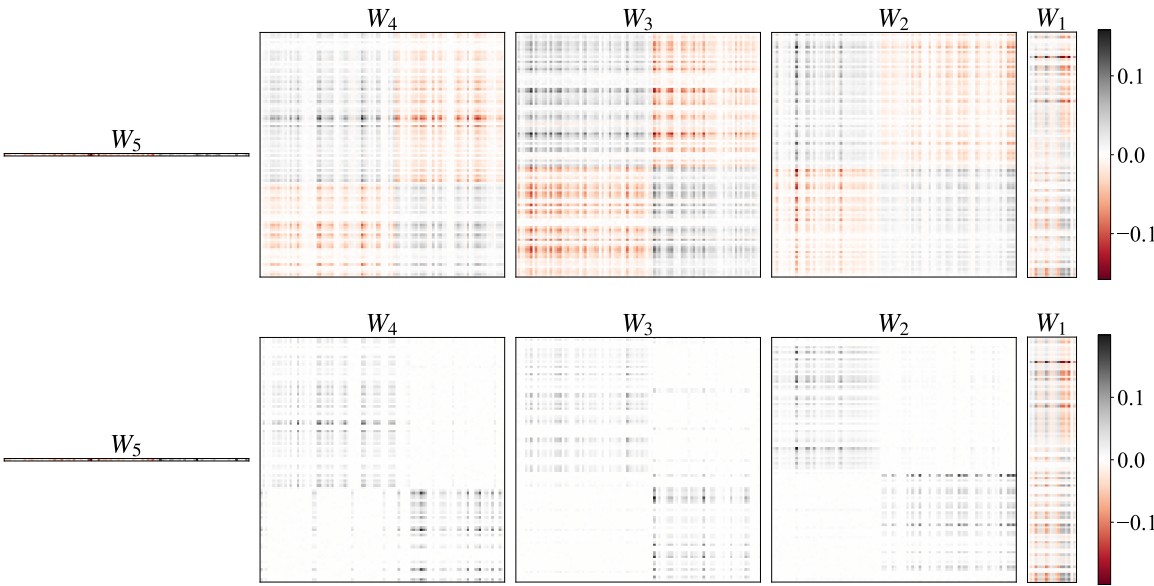

(b) Weights in a 5-layer linear (top) and ReLU (bottom) network as Equations (17) and (18).

Figure 10: Same as Figure 5 but with deeper networks.

Equation (19) is thus proven.

We now prove that under Equation (18) on the weights at time $t = t_0$, we have that $\forall\, t \geq t_0$, Equation (18) remains valid. We assume that $\sigma'(0) = 0$. We substitute the low-rank weights defined in Equation (18) into the learning dynamics of deep bias-free ReLU networks and make simplifications. For the first layer,

$$
\begin{aligned}
\tau \dot{\boldsymbol{W}}_1 &= \left\langle \sigma'(\boldsymbol{W}_1 \boldsymbol{x}) \odot \boldsymbol{W}_2^\top \cdots \boldsymbol{W}_L^\top (y - f(\boldsymbol{x})) \boldsymbol{x}^\top \right\rangle \\
&= \frac{u^{L-1}}{(\sqrt{2})^{L-2}} \left\langle \sigma'(\boldsymbol{W}_1 \boldsymbol{x}) \odot \boldsymbol{r}_1 \left( y - \left(\frac{u}{\sqrt{2}}\right)^L \boldsymbol{r}^\top \boldsymbol{x} \right) \boldsymbol{x}^\top \right\rangle \\
&= \frac{u^{L-1}}{(\sqrt{2})^L} \begin{bmatrix} \boldsymbol{r}_1^+ \\ \boldsymbol{0} \end{bmatrix} \left\langle \left( y - \left(\frac{u}{\sqrt{2}}\right)^L \boldsymbol{r}^\top \boldsymbol{x} \right) \boldsymbol{x}^\top \right\rangle_{\mathbb{S}^+} + \frac{u^{L-1}}{(\sqrt{2})^L} \begin{bmatrix} \boldsymbol{0} \\ \boldsymbol{r}_1^- \end{bmatrix} \left\langle \left( y - \left(\frac{u}{\sqrt{2}}\right)^L \boldsymbol{r}^\top \boldsymbol{x} \right) \boldsymbol{x}^\top \right\rangle_{\mathbb{S}^-} \\
&= \frac{u^{L-1}}{(\sqrt{2})^L} \boldsymbol{r}_1 \left( \boldsymbol{\beta}^\top - \left(\frac{u}{\sqrt{2}}\right)^L \boldsymbol{r}^\top \boldsymbol{\Sigma} \right).
\end{aligned}
\tag{55}
$$

For intermediate layers $1 < l < L$,

$$
\begin{aligned}
\tau \dot{\boldsymbol{W}}_l &= \left\langle \sigma'(\boldsymbol{h}_l) \odot \boldsymbol{W}_{l+1}^\top \cdots \boldsymbol{W}_L^\top (y - f(\boldsymbol{x})) \sigma(\boldsymbol{h}_{l-1})^\top \right\rangle \\
&= \left(\frac{u}{\sqrt{2}}\right)^{L-1} \begin{bmatrix} \boldsymbol{1} \\ \boldsymbol{0} \end{bmatrix} \odot \begin{bmatrix} \boldsymbol{r}_l^+ \\ \boldsymbol{r}_l^- \end{bmatrix} \left\langle \left( y - \left(\frac{u}{\sqrt{2}}\right)^L \boldsymbol{r}^\top \boldsymbol{x} \right) \boldsymbol{x}^\top \right\rangle_{\mathbb{S}^+} \boldsymbol{r} \begin{bmatrix} {\boldsymbol{r}_{l-1}^+}^\top & \boldsymbol{0} \end{bmatrix} \\
&\quad + \left(\frac{u}{\sqrt{2}}\right)^{L-1} \begin{bmatrix} \boldsymbol{0} \\ \boldsymbol{1} \end{bmatrix} \odot \begin{bmatrix} \boldsymbol{r}_l^+ \\ \boldsymbol{r}_l^- \end{bmatrix} \left\langle \left( y - \left(\frac{u}{\sqrt{2}}\right)^L \boldsymbol{r}^\top \boldsymbol{x} \right) \boldsymbol{x}^\top \right\rangle_{\mathbb{S}^-} \boldsymbol{r} \begin{bmatrix} \boldsymbol{0} & {\boldsymbol{r}_{l-1}^-}^\top \end{bmatrix} \\
&= \left(\frac{u}{\sqrt{2}}\right)^{L-1} \begin{bmatrix} \boldsymbol{r}_l^+ {\boldsymbol{r}_{l-1}^+}^\top & \boldsymbol{0} \\ \boldsymbol{0} & \boldsymbol{0} \end{bmatrix} \left( \boldsymbol{\beta}^\top - \left(\frac{u}{\sqrt{2}}\right)^L \boldsymbol{r}^\top \boldsymbol{\Sigma} \right) \boldsymbol{r} \\
&\quad + \left(\frac{u}{\sqrt{2}}\right)^{L-1} \begin{bmatrix} \boldsymbol{0} & \boldsymbol{0} \\ \boldsymbol{0} & \boldsymbol{r}_l^- {\boldsymbol{r}_{l-1}^-}^\top \end{bmatrix} \left( \boldsymbol{\beta}^\top - \left(\frac{u}{\sqrt{2}}\right)^L \boldsymbol{r}^\top \boldsymbol{\Sigma} \right) \boldsymbol{r} \\
&= \frac{u^{L-1}}{(\sqrt{2})^L} \begin{bmatrix} \sqrt{2}\,\boldsymbol{r}_l^+ {\boldsymbol{r}_{l-1}^+}^\top & \boldsymbol{0} \\ \boldsymbol{0} & \sqrt{2}\,\boldsymbol{r}_l^- {\boldsymbol{r}_{l-1}^-}^\top \end{bmatrix} \left( \boldsymbol{\beta}^\top - \left(\frac{u}{\sqrt{2}}\right)^L \boldsymbol{r}^\top \boldsymbol{\Sigma} \right) \boldsymbol{r}.
\end{aligned}
\tag{56}
$$

For the last layer,

$$
\begin{aligned}
\tau \dot{\boldsymbol{W}}_L &= \left\langle (y - f(\boldsymbol{x})) \sigma(\boldsymbol{h}_{L-1})^\top \right\rangle \\
&= \frac{u^{L-1}}{(\sqrt{2})^L} \left\langle \left( y - \left(\frac{u}{\sqrt{2}}\right)^L \boldsymbol{r}^\top \boldsymbol{x} \right) \boldsymbol{x}^\top \right\rangle_{\mathbb{S}^+} \boldsymbol{r} \begin{bmatrix} {\boldsymbol{r}_{L-1}^+}^\top & \boldsymbol{0} \end{bmatrix} \\
&\quad + \frac{u^{L-1}}{(\sqrt{2})^L} \left\langle \left( y - \left(\frac{u}{\sqrt{2}}\right)^L \boldsymbol{r}^\top \boldsymbol{x} \right) \boldsymbol{x}^\top \right\rangle_{\mathbb{S}^-} \boldsymbol{r} \begin{bmatrix} \boldsymbol{0} & {\boldsymbol{r}_{L-1}^-}^\top \end{bmatrix} \\
&= \frac{u^{L-1}}{(\sqrt{2})^L} \left( \boldsymbol{\beta}^\top - \left(\frac{u}{\sqrt{2}}\right)^L \boldsymbol{r}^\top \boldsymbol{\Sigma} \right) \boldsymbol{r} \boldsymbol{r}_{L-1}^\top.
\end{aligned}
\tag{57}
$$

Equations (55) to (57) can be rewritten as Equation (20) if we substitute the weights back in. The dynamics of the deep ReLU network as in Equation (20) is the same as a deep linear network as in Equation (7) except for constant coefficients.

We now prove that the low-rank weights remain low-rank once formed. We substitute the low-rank weights defined in Equation (18) into the left-hand side of Equations (55) to (57) and get

$$\tau \frac{d}{dt} u \boldsymbol{r}_1 \boldsymbol{r}^\top = \frac{u^{L-1}}{(\sqrt{2})^L} \boldsymbol{r}_1 \left( \boldsymbol{\beta}^\top - \left( \frac{u}{\sqrt{2}} \right)^L \boldsymbol{r}^\top \boldsymbol{\Sigma} \right),$$

$$\tau \frac{d}{dt} u \begin{bmatrix} \sqrt{2} \boldsymbol{r}_l^+ \boldsymbol{r}_{l-1}^{+\top} & \mathbf{0} \\ \mathbf{0} & \sqrt{2} \boldsymbol{r}_l^- \boldsymbol{r}_{l-1}^{-\top} \end{bmatrix} = \frac{u^{L-1}}{(\sqrt{2})^L} \begin{bmatrix} \sqrt{2} \boldsymbol{r}_l^+ \boldsymbol{r}_{l-1}^{+\top} & \mathbf{0} \\ \mathbf{0} & \sqrt{2} \boldsymbol{r}_l^- \boldsymbol{r}_{l-1}^{-\top} \end{bmatrix} \left( \boldsymbol{\beta}^\top - \left( \frac{u}{\sqrt{2}} \right)^L \boldsymbol{r}^\top \boldsymbol{\Sigma} \right) \boldsymbol{r},$$

$$\tau \frac{d}{dt} u \boldsymbol{r}_{L-1}^\top = \frac{u^{L-1}}{(\sqrt{2})^L} \left( \boldsymbol{\beta}^\top - \left( \frac{u}{\sqrt{2}} \right)^L \boldsymbol{r}^\top \boldsymbol{\Sigma} \right) \boldsymbol{r} \boldsymbol{r}_{L-1}^\top.$$

We cancel out the nonzero common terms on both sides and reduce the dynamics to two differential equations. The first one is about the norm of a layer $u$. The second one is about the rank-one direction in the first layer $u\boldsymbol{r}$.

$$\tau \frac{d}{dt} u = \frac{u^{L-1}}{(\sqrt{2})^L} \left( \boldsymbol{\beta}^\top - \left( \frac{u}{\sqrt{2}} \right)^L \boldsymbol{r}^\top \boldsymbol{\Sigma} \right) \boldsymbol{r},$$

$$\tau \frac{d}{dt} u \boldsymbol{r}^\top = \frac{u^{L-1}}{(\sqrt{2})^L} \left( \boldsymbol{\beta}^\top - \left( \frac{u}{\sqrt{2}} \right)^L \boldsymbol{r}^\top \boldsymbol{\Sigma} \right).$$

After the weights have formed the low-rank structure specified in Equation (18), the norm of each layer $u$ and the rank-one direction of the first layer $\boldsymbol{r}$ evolve while $\boldsymbol{r}_1, \boldsymbol{r}_2, \cdots, \boldsymbol{r}_{L-1}$ stay fixed. Hence, under Equation (18) on the weights at time $t = t_0$, we have that $\forall\, t \geq t_0$, Equation (18) remains valid. $\qquad\square$

We complement Figure 5 in the main text with Figure 10, which shows the converged weights in deeper bias-free linear and ReLU networks. Figures 5 and 10 are the empirical results that motivate us to make Conjecture 11.

## F  Depth Separation

A clarification on Section 3.2 is that deep bias-free ReLU networks are not more expressive than their two-layer counterparts if the input is scalar. For scalar input functions, the only positively homogeneous odd function is the linear function. Neither two-layer nor deep bias-free ReLU networks can express nonlinear odd functions with scalar input.

Another relevant fact is that there is also depth separation between two-layer and deep ReLU networks with bias. One example is the pyramid function, $\sigma(1 - \|\boldsymbol{x}\|_1)$, which was studied in (Ongie et al., 2020, Example 4) and (Nacson et al., 2023, Proposition 2).

## G   Additional Figure

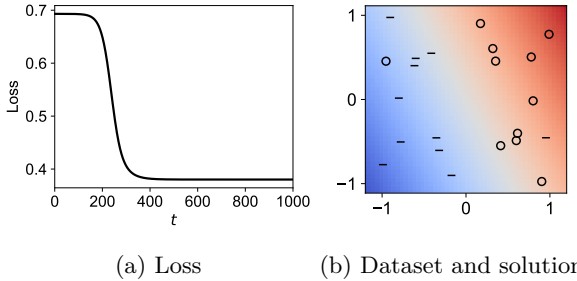

(a) Loss                          (b) Dataset and solution

Figure 11: Two-layer bias-free ReLU network trained on a linearly separable binary classification task with label flipping noise. Since the dataset satisfies symmetric Condition 3, the network follows linear network dynamics and converges to a linear decision boundary, which is a presumably robust solution here as it avoids overfitting the two noisy labels. (a) Loss curve. Logistic loss is used here. (b) The dataset is plotted with empty circles and short lines, representing data points with $+1$ labels and $-1$ labels respectively. The network output at the end of training is plotted in color.

## H   Implementation Details

All networks are initialized with small random weights. Specifically, the initial weights in the $l$-th layer are sampled i.i.d. from a normal distribution $\mathcal{N}(0, w_{\mathrm{init}}^2/N_l)$ where $N_l$ is the number of weight parameters in the $l$-th layer. The initialization scale $w_{\mathrm{init}}$ is specified below.

**Figure 1**. The networks have width 100. The initialization scale $w_{\mathrm{init}} = 10^{-2}$. The learning rate is 0.2. The two-layer networks are trained 10000 epochs. The three-layer networks are trained 80000 epochs. The dataset is plotted in the figure. The size of the datasets is 120.

**Figure 3**. The networks have width 500. The initialization scale is $w_{\mathrm{init}} = 10^{-8}$. The learning rate is 0.004. The input is 20-dimensional, $\boldsymbol{x} \in \mathbb{R}^{20}$. We sample 1000 i.i.d. vectors $\boldsymbol{x}_n \sim \mathcal{N}(\boldsymbol{0}, \boldsymbol{I})$ and include both $\boldsymbol{x}_n$ and $-\boldsymbol{x}_n$ in the dataset, resulting in 2000 data points. The output is generated as $y = \boldsymbol{w}^\top \boldsymbol{x} + \sin\left(4\boldsymbol{w}^\top \boldsymbol{x}\right)$ where elements of $\boldsymbol{w}$ are randomly sampled from a uniform distribution $\mathcal{U}[-0.5, 0.5]$. This dataset satisfy Condition 3 since the empirical input distribution is even and the output is generated by an odd function.

**Figures 4 and 8**. We use the same hyperparameters as Boursier et al. (2022). The network width is 60. The initialization scale $w_{\mathrm{init}} = 10^{-6}$. The learning rate is 0.001 for square loss and 0.004 for logistic loss. The orthogonal input dataset contains two data points, i.e., $[-0.5, 1], [2, 1]$. The XOR input dataset contains four data points, i.e., $[0, 1], [2, 0], [0, -3], [-4, 0]$.

**Figure 5**. The networks have width 100. The initialization scale $w_{\mathrm{init}} = 10^{-2}$. The learning rate is 0.1. The networks are trained 20000 epochs. The dataset is generated in the same way as Figure 3 except that the output is generated as $y = \boldsymbol{w}^\top \boldsymbol{x}$.

**Figure 6**. The network width is 100. The initialization scale $w_{\mathrm{init}} = 10^{-3}$. The learning rate is 0.025. The dataset contains six data points: $[1, 1], [-1, -1], [1, -1], [-1, 1], [-1, 0], [1, \delta]$.

