# OpenReview forum: "When Are Bias-Free ReLU Networks Effectively Linear Networks?"
_TMLR — Accepted by TMLR_

### Review · Reviewer_TNKe · 2025-02-07

**Summary Of Contributions:**

This paper studies the (theoretical) implication of omitting bias terms in the parametrization of (two layer) ReLU networks. It provides a short proof of the fact it can only represent the sum of a linear function and an even function in the case of 2 layer, as well as providing a depth separation example, where more complex odd functions can be represented.

It is also shown that under a specific set of assumptions on both the dataset and initialization, the training dynamics of two layer ReLU networks is equivalent to the training dynamics of a linear network. The authors then complement these theoretical results with experiments on orthogonal and XOR datasets, as well as deeper architectures.

**Audience:**

Yes

**Claims And Evidence:**

Yes

**Requested Changes:**

Following my concerns mentioned above, I wish the authors could provide the following changes to the current paper:

- add rigorous, mathematical definitions in the paper when needed
- clarify assumption 5: I am not sure yet of how restrictive/common it currently is
- add a figure illustrating the function g(x) (equation 9)

Lastly for for Theorem 1, I believe a stronger theorem with an equality of sets could be possible. I think for example following the same kind of analytical tools as in *How do infinite width bounded norm networks look in function space?* by Savarese et al (2019) might lead to the reverse inclusion.

**Strengths And Weaknesses:**

I think this paper suits well the line of TMLR. The proofs are simple to follow and I can thus assess their soundness. Additionally, the contribution brings nuance to previous literature on ReLU networks by showing that under some conditions, the studied dynamics is no different from a linear network ones.

My main concerns are about the clarity of the paper. First, some theoretical statements are given without any math definition, making the claimed results unclear at times. For example, I do not understand what does the second part of Assumption 5 means "W2 has equal L2 norms for its positive and negative elements". Similarly, Theorem 1 is only written in words. As a consequence, it kind of hides that the statement is only an inclusion (i.e., the set of functions that can be represented by neural networks is included in ...).
On a similar note, I did not understand what the sentence "To simplify the analysis, Assumption 5 assumes ..." means. Does this mean that $W_1$ is rank one at initalization? If so, this seems like a very strong assumption.
As a last example of such lack of definition, it is mentioned several times that "the target function is linear". What does this mean? I understand that as
$$y_i=x_i^{\top}w^{\star} +\varepsilon_i,$$
for some random noise $\varepsilon_i$. But here, it is not clear whether such a noise is included or not in the statement.

Maybe related to this point, I had a hard time to understand Section 5. I did not get all the notations, but also, it is here hard to parse what properties are just "empirical observations" (for what kind of init, what setup?) and mathematical derivations from the observations/assumptions. This gives a "speculative vibe", from which I have difficulties to extract interesting insights.
Also in this section, I did not understand why "which is true when the network width approaches infinity" is actually true. Some additional explanations might be helpful.

## Minor remarks

About the literature related work:
- I am unsure about the discussion of Soudry et al (2018) in Section 6. To me, convergence towards max-margin holds, whether or not we count any bias term. It's just that the definition of max-margin solution differs whether or not we count the bias norm in the norm penalization.
- *The convergence rate of neural networks for learned functions of different frequencies* by Ronen et al (2019) already studied the functions that can be represented by no-bias ReLU networks. Your proof and result is much simpler and clearer, but I think it would still be great to mention this paper somewhere.
- To complete the discussion on the perturbed symmetric dataset, the paper *Simplicity bias and optimization threshold in two-layer ReLU networks* by Boursier and Flammarion (2024) even seems to suggest that for finite values of $\Delta y$, the plateau you mention might even be infinite, i.e., the optimization scheme is unable to leave this plateau even for an infinite time (while you here suggest it might leave it after a duration that grows larger as $\Delta y\to 0$.

---

> ### Author Response · Authors · 2025-03-11
>
> We thank the reviewer for very prompt and constructive feedback. We have taken the suggestions to revise the paper. Our revisions and replies to questions are outlined below.
>
> - **Add Mathematical Definitions**
>
>   We reworded Theorem 1 to explicitly state the set inclusion.
>
>   We added $||\max(W_2,0)||=||\max(-W_2,0)||$ in Assumption 7 (Assumption 5 before revision), specifying what we mean by "the L2 norms of the positive and negative elements in $W_2$ are equal".
>
>   We have reorganized Section 5, separating it into a conjecture followed by a proposition conditional on the conjecture. We hope it clarifies that the conjecture is motivated by empirical observations, while the proposition conditional on it is a proven result. In our conjecture, we specify that a linear target function is $y={w^*}^\top x$, thus not including noise. We also added Figure 10 with deeper bias-free ReLU networks to further support our conjecture.
>
> - **Clarify Assumption 7** (Assumption 5 before revision)
>
>   Thank you for raising this important point. We have taken a suggestion from reviewer xhwY to move two lemmas from the Appendix to the main text, which helps us motivate Assumption 7.
>
>   In the revised version, Lemma 6 comes before Assumption 7. Lemma 6 shows that the weights initialized from small random values naturally evolve to be approximately rank-one with $O(w_{\text{init}})$ errors, which vanishes when the initialization scale $w_{\text{init}}$ approaches zero. Thus, the approximate rank-one weights are a natural consequence of training from small random weights, and the exact rank-one assumption is made for technical convenience.
>
>   Additionally, all of our experiments use small random initialization, which does not satisfy the rank-one weights assumption but reflects the realistic initialization scheme for rich learning. Nonetheless, the equivalence stated in Theorem 8 holds with <0.3% error in Figure 3b. Further, in Figure 7, we empirically show that the equivalence in Theorem 8 approximately holds even under large initialization and a large learning rate.
>
> - **Strengthen Theorem 1**
>
>   Thank you very much for sharing an idea to help us strengthen Theorem 1. The result in Savarese et al [1] should indeed apply to our bias-free network in the case of 1D input, $x\in \mathbb R$. However, the extension to multi-dimensional input is non-trivial, as studied in a follow-up paper by Savarese's coauthors [2]. Our understanding is that the reverse inclusion of Theorem 1 might not hold, because positively homogeneous functions can be neither piece-wise linear nor smooth in directions orthogonal to radial lines, and consequently cannot be expressed by two-layer bias-free ReLU networks with bounded norm of weights.
>
>   We welcome any feedback or correction from the reviewer -- we really appreciate your expertise on this topic.
>
> - Our figure illustrating $g(x)$ has been moved from the Appendix to the main text just beside the definition of $g(x)$.
>
> [1] Savarese, P., Evron, I., Soudry, D., & Srebro, N. How do infinite width bounded norm networks look in function space? COLT 2019.
>
> [2] Ongie, G., Willett, R., Soudry, D., & Srebro, N. A Function Space View of Bounded Norm Infinite Width ReLU Nets: The Multivariate Case. ICLR 2020.
>
> **Related Work**
>
> - We agree with your understanding of Soudry et al (2018): the difference between networks with or without bias is whether the bias norm is penalized or not. This was discussed by Soudry et al (2018) in their paragraph titled "non-homogeneous linear predictors". The hard-margin max-margin solution is defined as:
>   $$
>   \underset{\mathbf w,b}{\text{argmin }}\|\mathbf w\|^2 \text{ s.t. } y_i(\mathbf w^\top \mathbf x_i + b) \geq 1 \tag{1}
>   $$
>   However, the network with bias converges (directionally) to a slightly different solution:
>   $$
>   \underset{\mathbf w,b}{\text{argmin }}(\|\mathbf w\|^2+b^2) \text{ s.t. } y_i(\mathbf w^\top \mathbf x_i + b) \geq 1 \tag{2}
>   $$
>   We intend to explain that the solution of a network with bias (Eqn. 2), is not a max-margin solution (Eqn. 1). This is because the objective in Eqn. 2 penalizes the bias term, leading to a preference for hyperplanes closer to the origin rather than being determined solely by margin maximization. We have reworded the paragraph in Section 6.1 to clarify this point -- please let us know if it's still misleading.
>
> - We have indeed discussed Ronen Basri's work in the first paragraph of our related work section, where it appears as Basri et al. (2019). Thank you for this note.
>
> - Thank you for pointing us to the related work by Boursier & Flammarion (2024). We have added a discussion of this paper in our Section 6.2 on 'Perturbed Symmetric Dataset'.

---

> > ### Comment · Reviewer_TNKe · 2025-03-12
> >
> > I thank the authors for their answer and edits. The paper now seems clearer to me. I agree with reviewer gZN1 that Assumption 7 (previously 5) is very strong and the limit as $w_{\mathrm{init}}\to 0$ is not enough to justify it. However, I feel it is now extensively discussed by the authors and somehow validated by the experiments.
> >
> > I thus find this work, in its current state, suitable for TMLR.

---

### Review · Reviewer_xhwY · 2025-02-07

**Summary Of Contributions:**

This manuscript studies conditions under which bias-free ReLU neural networks display identical gradient flow training dynamics to linear networks. Its main result (Theorem 7) shows an equivalence for networks with a single hidden layer under symmetry assumptions on the data and an alignment assumption on the weights. The remainder of the paper consideres extensions to deeper networks and other datasets through a combination of partial analytical results and numerics.

**Audience:**

Yes

**Broader Impact Concerns:**

I do not see any broader impact concerns.

**Claims And Evidence:**

Yes

**Requested Changes:**

Please note that I've ordered these requested changes by appearance in the text, not importance.

1. Though the proof of Theorem 1 is immediate, I think it would be clearer to first note that the leaky ReLU function admits a decomposition as $\max(x,\alpha x) = \frac{1+\alpha}{2} x + \frac{1-\alpha}{2} |x|$ and then that this leads to a decomposition of the network as a whole. This decomposition is implied by the drawing below equation (8), but I think this schematic is less helpful than simply stating the decomposition of the activation function.

2. I think the clarity of Section 4 could be improved by deferring less material to the appendices, and better referencing what is shown there. First, Remark 6 should be made more mathematically precise. I would suggest that the authors state Theorem 15 from Appendix C in the main text, and also give a precise statement of the conditions under which the assumption that "the second-layer weights have equal L2 norm for their positive and negative elements [...] is justified by the width approaching infinity". Next, the authors should state below Theorem 7 that its proof can be found in Appendix C.2, rather than only referencing that appendix in a parenthetical above. Also, is there a reason why Equations (10a-b) are stated separately from Theorem 7, rather than as part of the Theorem?

3. It would be useful to state Lemma 11 in the main text, as this is fundamentally why the authors arrive at Condition 3. Indeed, the proof of Theorem 7 depends on Condition 3 only through Lemma 11 (up to replacement of the factors of 1/2 in the derivation of equation 36).

4. Is Corollary 9 correct as written? As I recall, Soudry et al. (2018) show that the norm of the weight vector continues to grow through training, and the convergence to the max-margin solution is only in direction. Please correct me if I've misremembered.

5. The desired take-away from Section 4.2 could be made more clear - is the idea here just to illustrate that with other datasets the equivalence with a single linear network breaks down but there can be more general equivalences?

6. It could be nice to highlight the main empirical results of Section 5 (i.e. equation 15) as a conjecture. Then, you could state equations (16) and (17) as a result conditional on that conjecture. Again, it would be useful to give a more nuanced statement of when the balancing of positive and negative parts of the weights hold at large width, rather than simply asserting that this is true.

**Strengths And Weaknesses:**

I think this paper should be of interest to the TMLR audience, as it collects a number of extensions to previous linearization results. The manuscript is relatively clearly written, but (as noted below under Requested Changes) there are places where I think its organization could be improved. The analysis is in general straightforward and the results appear mathematically correct, though I have not exhaustively checked the proofs.

---

> ### Author Response · Authors · 2025-03-11
>
> We thank the reviewer for their prompt review and offering very clear and thoughtful suggestions. We have revised our paper as suggested and would like to outline our changes below. The numbers in our list match the reviewer's list.
>
> 1. We have revised the proof for Theorem 1 to start with the decomposition: $\max(z,\alpha z)=\frac{1+\alpha}2z + \frac{1-\alpha}2z$.
> 2. We have moved Lemma 6 (Theorem 15 before revision) from Appendix to the main text. We also moved Equation 12 (Equation 10 before revision) to be a part of the main Theorem 8. We refined the cross-references, pointing to the proof of a theorem immediately after it. Thank you for these suggestions -- they indeed help us motivate our initialization assumption and explain the main Theorem.
> 3. We have moved Lemma 5 (Lemma 11 before revision) from Appendix to the main text. You're right that it's the key implication of Condition 3 that we exploit to prove our main theorem.
> 4. Thank you for your careful attention here. With logistic loss, the norm of weights indeed diverges and the convergence to max-margin solution is only in direction. We have corrected this point in Corollary 10. The directional convergence is now specified as: $w^* / || w^* || = w_{\mathrm{svm}} / || w_{\mathrm{svm}} ||$.
> 5. We have revised Section 4.2 to clarify our intended take-away is that: for more general datasets, a ReLU network does not have equivalent dynamics as a linear network, but comparing ReLU with linear networks remains useful for understanding the learning dynamics of ReLU networks.
> 6. We have reorganized Section 5 to present a conjecture followed by a proposition conditional on it. It does help make the distinction between conjectured and proved results in Section 5 clearer.

---

> > ### Comment · Reviewer_xhwY · 2025-03-11
> >
> > I've looked through the authors' responses to my comments and those of the other reviewers, and I think the concerns have adequately been addressed. My opinion remains that this paper is very suitable for publication in TMLR.

---

### Review · Reviewer_gZN1 · 2025-02-27

**Summary Of Contributions:**

This paper studies the expressivity and training dynamics of two-layer and deep bias-free ReLU networks, and identifies conditions under which they are equivalent to linear networks. In particular, under the assumption that the input distribution is even and the label is an odd function of the input, the authors prove that when using a particular rank-one initialization for the first layer, the gradient flow dynamics remains identical to a time-rescaled dynamics for linear networks. The authors also demonstrate that on orthogonal datasets, the dynamics of two-layer bias-free ReLU networks can be identified with multiple linear networks. Finally, the authors observe certain sutrcutres in deep bias-free ReLU networks that are similar to those of deep linear networks, leading to equivalent training dynamics in specific settings.

**Audience:**

Yes

**Broader Impact Concerns:**

Not applicable.

**Claims And Evidence:**

Yes

**Requested Changes:**

* I believe to achieve the rich training regime, one can initialize the coordinates of $W_2$ to be of order $1/H$, while the rows of $W_1$ can be unit-norm (therefore not necessarily) and are typically initialized isotropically in the space, see e.g. Table 1 of Yang and Hu, 2020. This would not correspond to ``small initialization'' for $W_1$. Can the authors show that with this type of initialization instead of Assumption 5, a statement similar to Theorem 7 holds, perhaps as $H \to \infty$?

* If the above scaling does not work, can the authors provide another initialization scaling which achieves the rich regime, is isotropic in the first layer, and for which a variant of Theorem 7 can be proved?

G. Yang and E. J. Hu. "Feature Learning in Infinite-Width Neural Networks." ICML 2021.

**Strengths And Weaknesses:**

**Strengths**: The study of the training dynamics of two-layer and deep ReLU networks beyond the lazy regime is challenging, with many aspects that remain unresolved. This paper provides a clean analysis of the optimization dynamics under particular assumptions in a bias-free setting, and is able to highlight the importance of the bias term to obtain highly expressive prediction functions.

**Weaknesses**:
* A key concern for me is Assumption 5, a core assumption in the theoretical results, which seems unrealistic. Specifically, it assumes a rank-1 initialization which is not used in practice. The authors argue that this is not an issue since the rich training regime uses small initialization, but then this assumption can be rewritten to better match the rich training regime and practical initializations. This is further discussed in the section below.

* There are not sufficient experiments in the paper to support the conjecture of Equation (15b). To my understanding the only empirical evidence is Figure 4 which considers a 3-layer ReLU network. More experiments are needed to understand the effect of depth and data distribution on this observation. This is the reason I answered "No" to Claims and Evidence, and I would be happy to change to "Yes" if the authors could point at additional experiments and justifications, or if they limit this section to networks of depth 3.

* The experiments in Section 4.2 seem to cover simple toy data distributions and two-layer ReLU networks. There is some theoretical justification of the observations in Appendix D. I think extending this theory to more than two samples and presenting precise theoretical results in the main text should be within reach, and would make this section much stronger.

---

> ### Author Response · Authors · 2025-03-11
>
> We thank the reviewer for their constructive suggestions and questions. We have revised our paper according to your suggestions, and would like to outline our revision below.
>
> - **Motivation for Initialization Assumption 7** (Assumption 5 before revision)
>
>   Thank you for raising this important point. In fact, we do consider the small isotropic initialization regime in our work.
>
>   We assume rank-one weights in Assumption 7 primarily for ease of technical derivations. However, we show that with small isotropic initialization, the weights will naturally evolve to be approximately rank-one with $O(w_{\text{init}})$ errors, which vanishes when the initialization scale $w_{\text{init}}$ approaches zero. This ' rank-one alignment from small initialization' behavior is formally stated in Lemma 6, which has been moved from Appendix to the main text, following a suggestion from reviewer xhwY. We have also revised the text around Lemma 6 and Assumption 7 to clarify that: the approximate rank-one alignment is a natural consequence of training from small isotropic random weights, and that the exact rank-one assumption is made only for technical tractability.
>
>   Additionally, all of our experiments use (isotropic) Gaussian initialization, which does not satisfy the rank-one weights assumption but reflects the realistic initialization scheme for rich learning. Nonetheless, the equivalence stated in Theorem 8 holds with <0.3% error in Figure 3b, where small isotropic initialization is used. Further, in Figure 7, we empirically show that the equivalence in Theorem 8 approximately holds even under large isotropic initialization and a large learning rate.
>
>   We also tried the initialization in Yang & Hu (2021) and didn't find an equivalence to linear networks there. However, we hope our revision now clarifies that our analysis handles the rich learning regime with isotropic small initialization, and Assumption 7 is supported by Lemma 6.
>
> - **More Experiments with Deep ReLU Networks**
>
>   Thank you for this constructive feedback. We have added Figure 10, which shows the converged weights in 4-layer and 5-layer bias-free ReLU networks. In bias-free ReLU networks with depth 3,4, and 5, we all empirically observe the low-rank weights that we conjectured in Equation 18 (Equation 15 before revision).
>
>   Also, we have reorganized Section 5, separating it into a conjecture followed by a proposition conditional on the conjecture. We hope it is now clearer that the conjecture is motivated by empirical observations, while the proposition conditional on it is proved.
>
> - **Extension of Section 4.2**
>
>   We have refined our derivations for Section 4.2 in Appendix D, where we formally state Assumption 17 about orthogonal datasets and use it to prove Proposition 18. Proposition 18 handles an arbitrary number of orthonormal inputs. We also add Figure 9 to present an experiment on an orthogonal dataset with more than two samples.
>
>   We have revised Section 4.2 to clarify that the connection to multiple linear networks applies to datasets with any number of orthogonal inputs. The reason for using a dataset with two samples in Figure 4 is to allow us to plot the data points and the first-layer weights.
>
>   If the reviewer thinks it's helpful, we can also move Proposition 18 from Appendix D to Section 4.2.
>
> We hope our revision is beneficial and welcome any further feedback.

---

> > ### Comment · Reviewer_gZN1 · 2025-04-04
> >
> > Thank you for your clarifications, my concerns are mostly resolved and I think the paper is suitable for the TMLR community.

---

### Decision · Action_Editor_EH6Q · 2025-04-13

**Recommendation:** Accept as is

**Comment:**

Based on the reviews, the manuscript provides an interesting contribution to the understanding of neural network dynamics, particularly highlighting the conditions under which bias-free ReLU networks exhibit linear behaviour. All three reviewers recognised the technical soundness of the main theoretical results and relevance to ongoing research beyond the lazy training regime.

The main concern raised by the reviewers was that the initialization assumption (Assumption 5) is restrictive and not standard in practice. The reviewers also made some constructive suggestions that were implemented by the authors, helping improving the original version of the manuscript.

**Audience:**

This paper is relevant to the audience of TMLR interested in machine learning theory.

**Claims And Evidence:**

The paper offers a theoretical analysis of conditions under which bias-free ReLU networks exhibit training dynamics similar to linear networks. The main claims are mathematically well supported, though some results are presented informally. The empirical results are consistent with the theoretical perspective and help illustrate the phenomenology.